**Data availability statement:** The data of YLBS in the manuscript is available in References [10] and can be requested from State Key Laboratory

# Enhanced intelligent train operation algorithms for metro train based on expert system and deep reinforcement learning

Yunhu Huang[1], Wenzhu Lai[2], Dewang Chen[3,4*], Geng Lin[1*], Jiateng Yin[5*]

**1** College of Computer and Data Science, Minjiang University, Fuzhou, Fujian, China, **2** Digital Banking Laboratory of Industrial and Commercial Bank of China Software Development Center, Zhuhai, Guangdong, China, **3** School of Transportation, Fujian University of Technology, Fuzhou, Fujian, China, **4** Fujian BeiDou Navigation and Intelligent Transportation Collaborative Innovation Center, Fuzhou, Fujian, China, **5** State Key Laboratory of Rail Traffic Control and Safety, Beijing Jiaotong University, Beijing, China

☯ These authors contributed equally to this work.

* dwchen@fjut.edu.cn, lingeng413@163.com, 13111054@bjtu.edu.cn, yunhuhuang@aliyun.com

## Abstract

In recent decades, automatic train operation (ATO) systems have been gradually adopted by many metro systems, primarily due to their cost-effectiveness and practicality. However, a critical examination reveals computational constraints, adaptability to unforeseen conditions and multi-objective balancing that our research aims to address. In this paper, expert knowledge is combined with deep reinforcement learning algorithm (Proximal Policy Optimization, PPO) and two enhanced intelligent train operation algorithms (EITO) are proposed. The first algorithm, $EITO_E$, is based on an expert system containing expert rules and a heuristic expert inference method. On the basis of $EITO_E$, we propose $EITO_P$ algorithm using the PPO algorithm to optimize multiple objectives by designing reinforcement learning strategies, rewards, and value functions. We also develop the double minimal-time distribution (DMTD) calculation method in the EITO implementation to achieve longer coasting distances and further optimize the energy consumption. Compared with previous works, EITO enables the control of continuous train operation without reference to offline speed profiles and optimizes several key performance indicators online. Finally, we conducted comparative tests of the manual driving, intelligent driving algorithm (ITOR, STON), and the algorithms proposed in this paper, EITO, using real line data from the Yizhuang Line of Beijing Metro (YLBS). The test results show that the EITO outperform the current intelligent driving algorithms and manual driving in terms of energy consumption and passengers' comfort. In addition, we further validated the robustness of EITO by selecting some complex lines with speed limits, gradients and different running times for testing on the YLBS. Overall, the $EITO_P$ algorithm has the best performance.

of Rail Traffic Control and Safety, Beijing
Jiaotong University (13111054@bjtu.edu.cn).

**Funding:** Funders: Fujian Provincial Education
and Scientific Research Project for Young and
Middle-aged Teachers (Science and Technology
Category) (JAT231096), The Natural Science
Foundation of Fujian Province (2024J011180,
2023J05251, 2024J08275, 2022J01117),
Innovation Star Talent Program of the Third
Batch in Fujian Province (003002), The Fujian
Provincial Social Science Foundation
(FJ2024C196), Minjiang University Talent
Introduction Research Project (MJU24003,
MJY23035), Special Fund for Education and
Scientific Research of Fujian Provincial
Department of Finance (GY-Z21001), Scientific
Research Foundation of Fujian University of
Technology (GY-Z22071). The funders provided
some ideas and directions in the study design,
and had role in data collection and analysis,
decision to publish, or preparation of the
manuscript. The specifications are as follows:
Funders: Fujian Province's Education Research
Project for Young and Middle-aged Teachers
under Grant JAT231096, Innovation Star Talent
Program of the Third Batch in Fujian Province
under Grant 003002, The Natural Science
Foundation of Fujian Province (2024J011180,
2023J05251, 2024J08275, 2022J01117),
Minjiang University Talent Introduction
Research Project under Grant MJU24003
provided financial support. Funders: Scientific
Research Foundation of Fujian University of
Technology under Grant GY-Z22071, Minjiang
University Talent Introduction Research Project
under Grant MJY23035, Science and Education
Joint Special Project of Minjiang University
(Science and Engineering Category) under Grant
MJKJ24005 provided support in the areas of
data collection and analysis. Funders: Special
Fund for Education and Scientific Research of
Fujian Provincial Department of Finance under
Grant GY-Z21001, and 2023 National College
Students' Innovation and Entrepreneurship
Training Program under Grant 202310395011
provided support in the preparation of the
manuscript.

**Competing interests:** The authors have
declared that no competing interests exist.

# 1 Introduction

In recent years, with the acceleration of urbanization, urban road traffic resources can not
meet the growing traffic demand, therefore, Intelligent Transportation System (ITS) [1] came
into being. The automatic train operation (ATO) system [2], which replaces manual driving
in many places with low cost and automation, has become an important part of ITS. While
ATO systems have been increasingly embraced by many metro systems over the past decades
due to their low cost and practicality, it is evident that they fall short in several critical areas.
Firstly, the intelligence of these systems is limited; they often rely on predefined operational
strategies and lack the dynamic adaptability to respond effectively to complex and unforeseen
circumstances. Secondly, the absence of self-learning capabilities restricts their potential to
improve efficiency and safety over time through the accumulation and analysis of operational
data. Lastly, the generalization of these systems is constrained; they are typically tailored to
specific lines and struggle to adapt to diverse line conditions, such as varying speed limits,
gradients, and traffic flows, which limits their broader application. These limitations under-
score the need for more advanced, intelligent, and adaptable train operation systems that can
enhance operational efficiency, safety, and passenger comfort.

The speed control of metro train operation can be represented as a multi-objective opti-
mization problem with constraints. In order to satisfy these constraints and optimize the
objectives, the train must make driving decisions based on real-time information. Under nor-
mal conditions, the ATO is responsible for all train traction and braking control commands
to make the train run on time, regulate its speed and stop exactly at its destination [3]. ATO is
traditionally divided into two sub-modules. The first one is dedicated to the calculation of the
speed profile of the future train operation. Under this module, offline optimization algorithms
are used to calculate the optimal speed profile in terms of performance and energy consump-
tion. The second sub-module works mainly to ensure that the train accurately tracks the given
speed profile.

Recently, many studies have been devoted to designing an offline optimized train trajec-
tory to improve energy efficiency. For example, Khmelnitsky [4] devised a numerical algo-
rithm to get the best velocity profile, taking into account changeable gradients and arbitrary
speed limits. Furthermore, train operation issues encompass a variety of additional factors,
such as trip comfort and punctuality. Yang et al. [5] created a genetic algorithm based on
binary coding, developing a two-target integer programming model with headway time con-
trol and dwell time management to find the optimal solution in terms of energy savings and
service quality. Wang et al. [6] introduced a new iterative convex planning (ICP) technique
to solve the train scheduling problem to achieve the ideal departure time, running time, and
dwell time in order to minimize travel time and energy consumption. Using optimal speed
trajectory searching methodologies under diverse track parameters, Guan et al. [7] created a
multi-objective optimization model for the speed trajectory, with energy consumption and
travel time as the key optimization objectives. With the development of artificial intelligence,
many intelligent algorithms have been applied to train operation. Akba et al. [8] employ an
artificial neural network with the genetic algorithm to optimize the coasting points of the
velocity-distance trajectory to obtain minimum energy expenditure for a given travel time.
Yang et al. [9] combined a simulation-based approach and a genetic algorithm to find an
approximate optimal coasting control strategy. Yin et al. [10] developed ITOR algorithm for
intelligent train operating capable of satisfying multiple objectives by using expert experience
and Q-Learning algorithm. Zhang et al. [11] used manual driving data to train (K-NN, Bag-
ging CART, and Adaboost CART) three well-known algorithms to predict the driver's out-
put control. Recently, Zhou et al. [12] proposed STO algorithm by using deep deterministic

policy gradient (DDPG) and normalized dominance function (NAF) algorithms to further optimize the energy consumption, comfort during train operation metrics.

After generating the optimal recommended speed profile, the ATO's task is to develop an efficient method to control the train relatively to different train models and operating conditions (e.g., tunnels, curves, steep gradients) so that the train can accurately track the speed profile and operate safely and smoothly. Ke et al. [13] proposed a fuzzy PID gain method to track the recommended speed profile, which was optimally generated by the MAX-MIN ant system. Song et al. [14] investigated at the consequences of time-varying failures in both the traction and braking phases of the train, and suggested an adaptive backstepping control system that was completely parameter-dependent and successful in achieving good speed tracking performance. Liu et al. [15] proposed a high-speed railway control system based on fuzzy control method and designed the control system in MATLAB. Gu et al. [16] have proposed a new energy-efficient train operation model based on real-time traffic information from a geometric and topographic perspective. Two robust adaptive control approaches considering actuator saturation and unknown system parameters were proposed by Gao et al. [17]. Recently, Pu et al. [18] proposed a model-free adaptive speed controller based on neural network (NN) and PID algorithms,and the effectiveness of the proposed algorithms to track the SD trajectory precisely is proved by numerical experiments and real-line applications.

Actually, the previous research and application has greatly improved the operational performance of metro train operation. However, there are still some basic problems that have not been solved, which hinder the development of ATO systems. Firstly, most existing ATO systems achieve their train operation goals by focusing on energy-efficient trajectory calculation, real-time tracking methods, and station parking algorithms, respectively. Especially, ATO algorithms are designed to track offline optimized speed profiles, lacking intelligence, flexibility and robustness. Few studies have comprehensively considered multiple objectives such as driving comfort, punctuality, parking accuracy, and energy consumption. Meanwhile, complex control methods are difficult to implement in real operation when faced with system non-linearity, unknown resistance and variable in-train forces. Secondly, modern metro trains are capable of outputting continuous traction and braking forces, but few studies have been conducted to design continuous control models considering complex line conditions, for example, the intelligent train operation algorithms based on reinforcement learning (ITOR) are proposed in [10], which can only achieve discrete control of train with simple line condition. Finally, there are some metro sections with more complex speed limits and gradients change metro, and most of the train models proposed only consider the operation in the intervals with simple speed limits and gradients conditions, such as the smart train operation (STO) algorithms based on normalized advantage function (STON) proposed in [12] is difficult to be applied to the case of long distances between two consecutive stations and lines with complex speed limits.

Facing these problems, new intelligent driving algorithms with a higher level of intelligence need to be investigated, which are called enhanced intelligent operation algorithms (EITO$_E$ and EITO$_P$) in this paper. On the one hand, experienced drivers combined with their long-term accumulated maneuvering experience can implement effective control of the train in real-time so that the train operation meets the requirements of several control objectives. Besides, they can be well adapted to different conditions of railroad lines. On the other hand, reinforcement learning(RL) has been used as a powerful decision tool [19] to tackle optimal control problems in many domains, such as micro-drone control [20], robot control [21], and with good results in the field of intelligent driving of trains [10], [12]. Meanwhile, deep reinforcement learning [22] is considered to be useful for the control of continuous movements [23], the detailed demonstration is analyzed in Sect 3.2.

Therefore, we consider combining expert (experienced drivers) experience with deep reinforcement learning algorithms to achieve better and intelligent operations. The necessity of proposing both (EITO$_E$ and EITO$_P$) lies in their complementary strengths. (EITO$_E$ leverages expert knowledge and heuristic rules to provide a robust baseline for intelligent train operation, while EITO$_P$) uses deep reinforcement learning (PPO) to optimize multiple objectives dynamically. By presenting both algorithms, we aim to demonstrate how expert knowledge can be effectively integrated with advanced machine learning techniques to enhance train operation performance. This dual approach allows for a comprehensive evaluation of their effectiveness under varying conditions, showcasing the versatility and adaptability of our proposed solutions. As can be seen from the above analysis, the contributions of this paper are as follows:

1) Integration of Expert Knowledge with Deep Reinforcement Learning: We introduce a novel approach that integrates expert system-based rules, distilled from experienced drivers, with the Proximal Policy Optimization (PPO) algorithm. This integration results in the development of EITO$_E$ and EITO$_P$ algorithms, which not only provide a robust operational baseline for intelligent train operation but also dynamically optimize multiple objectives, enhancing the adaptability and efficiency of train control systems.

2) Development of EITO$_E$ Algorithm: The EITO$_E$ algorithm is developed by encapsulating heuristic rules and inference methods from expert drivers within an expert system framework. This innovation allows for the generation of control strategies independent of offline speed profiles, thereby offering a flexible and adaptive operational approach that is responsive to real-time train operation requirements.

3) EITO$_P$ Algorithm for Multi-Objective Optimization: Extending the capabilities of EITO$_E$, the EITO$_P$ algorithm utilizes PPO to optimize key operational objectives including safety, punctuality, energy efficiency, and passenger comfort. A significant contribution of EITO$_P$ is its real-time adjustment of acceleration and braking strategies based on current train conditions and speed limits, which is crucial for maintaining energy efficiency and punctuality in metro train operations.

The rest of the paper is organized as follows. In Sect 2, we define the necessary mathematical notation and performance indicators for metro train operation, and then, we describe the problem of metro train operation. Sect 3 presents the design of the EITO algorithm based on the expert system and PPO. In Sect 4, we construct an EITO simulation platform and give three numerical examples of real data from YLBS. We conclude the paper in Sect 5.

## 2 Problem formulation and objectives

### 2.1 Problem statement

This section first formulates the train operation problem and then clearly states the objectives that the proposed algorithms aim to achieve.

The train control problem is formulated as an optimal control problem, focusing on finding an optimal control strategy for the traction and braking force during the travel time. First, the minimum time interval and the travel time of trains are defined as Eqs (1) and (2) respectively:

$$t_{i+1} = t_i + \Delta t. \tag{1}$$

For $0 \leq i \leq n - 1$, total travel time $T$ is defined as:

$$T = t_n - t_0. \tag{2}$$

where the initial run time is $t_0 = 0$ $(s)$, and the minimum time interval is $\Delta t = 0.02$ $(s)$.

The train motion model, which incorporates a multi - point mass and signal coordinate model, is used to simulate the electric multiple unit (EMU) of the train. This model takes into account the interaction effects between vehicles, offering an advantage over the traditional single - point train model. The model is expressed as:

$$Mu = (\lambda_1, \lambda_2, \lambda_3, \cdots, \lambda_n) \begin{pmatrix} F_1 \\ F_2 \\ F_3 \\ \vdots \\ F_n \end{pmatrix} - f - f_d. \tag{3}$$

where $M = \sum_{i=1}^{n} m_i$ denotes the weight of the EMU and $m_i$ denotes the weight of the i-th vehicle. $f_d = \sum_{i=1}^{n-1} \left( \Delta \ddot{l}_i \sum_{j=i+1}^{n} m_j \right)$ denotes the interaction between vehicle [24]. $F_1, F_2, F_3, \cdots, F_n$ denotes the force of each vehicle in the moving train, and $\lambda_i$ is the distribution constant that determines the acceleration/braking force of the i-th vehicle. $\Delta \ddot{l}$ denotes the variation of the spring deformation of the coupler. $f = f_r + f_g + f_c$ denotes the drag force. In addition, $f_r = \alpha v^2 + \beta v + \gamma$ describes the drag force caused by friction, The $\alpha, \beta, \gamma$ are the vehicle-specific factor. $f_c$ is the curve drag force defined as $f_c = 6.3M/r(s) - 55$, and $r(s)$ is the radius of the curve [11]. $f_g = Mg \sin \alpha$ is the drag force caused by the gradient, and $\alpha$ is the gradients angle.

The multi-unit model in Eq (3) captures inter-vehicle dynamics (e.g., coupler forces, mass distribution) to better simulate real-world EMUs. While the control force $u$ is centralized, its distribution across vehicles is governed by the force allocation constants $\lambda_i$ (Sect 2.1). For simplicity, we assumed uniform distribution ($\lambda_i = 1/n$) in simulations, as fine-grained force allocation is hardware-dependent and beyond this paper's scope.

Furthermore, the train acceleration (or braking) system in the study has nonlinear and time delays. Transfer function of the simulated brake acceleration system:

$$G(s) = \frac{\alpha_0}{1 + \tau s} e^{-\sigma s}. \tag{4}$$

where $G(s)$ means the actual output, $\alpha_0$ is the system performance gain, $\tau$ and $\sigma$ means the delay and time constant of the train acceleration/braking model, respectively.

Metro train operation control models are generally evaluated in terms of five aspects: safety, punctuality, energy consumption, passenger comfort, and parking accuracy.

- *Safety*: There may be multiple speed limit points between two consecutive metro stations, as shown in Fig 1, where $V_1^{\text{limit}}$, $V_2^{\text{limit}}$, $V_3^{\text{limit}}$, and $V_4^{\text{limit}}$ are the speed limits for different sections between the two stations. That means during the travel period, the speed of the train must be lower than the current speed limit of the railroad section to ensure safety. The safety evaluation index $I_s$ is defined as:

$$I_s = \begin{cases} 1 & v_i \le V_1^{\text{limit}} \; (\forall \; i) \\ 0 & v_i > V_1^{\text{limit}} \; (\exists \; i) \end{cases} \tag{5}$$

It is note that the intention of Eq (5) is to ensure that the train's speed remains within the designated limits. To explicitly state that the evaluation index $I_s$ is designed to enforce speed limits to prevent any misinterpretation. The condition should ensure that if the speed exceeds the limit, the index will reflect a violation, thereby discouraging overspeeding.

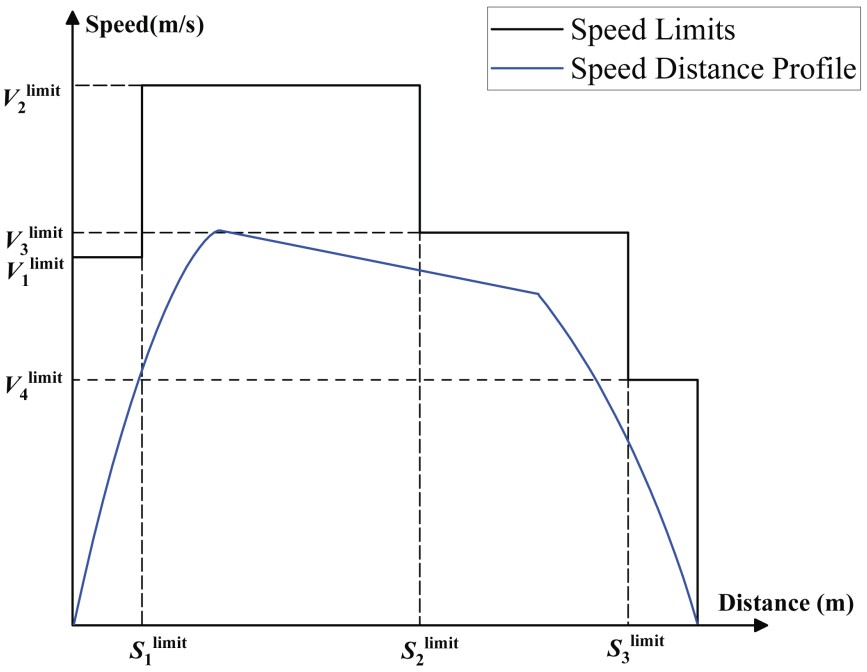

**Fig 1. Speed limits.**

- *Punctuality*: Punctuality is an important indicator of metro train operation that affects passenger interchanges and the entire schedule. We first define the running time error as:

$$e_t = |T_p - T_a| \tag{6}$$

where $T_a$ is the actual running time of the train and $T_p$ is the planned trip time of the train. In this paper, if the running time error is greater than 3 s, the metro is not running on time. Therefore, the punctuality evaluation index $I_t$ is defined as:

$$I_t = \begin{cases} 1 & e_t \leq 3 \\ 0 & e_t > 3 \end{cases} \tag{7}$$

- *Energy efficiency*: Energy consumption accounts for a large portion of train operating costs. The energy consumed is described as

$$E = \sum_{i=1}^{n} \left( M|u_i|v(t_i)\,\Delta t \right). \tag{8}$$

and the unit mass energy efficiency evaluation index between the two stations is defined as :

$$I_\varepsilon = \frac{E}{M}. \tag{9}$$

- *Comfort*: Comfort is a direct evaluation criterion for the quality of train service, which ensures that the instantaneous change in acceleration or deceleration should be below a

certain threshold value. We define the rate of acceleration $u$ change as:

$$\Delta u_i = \left|\frac{u_i - u_{i-1}}{\Delta t}\right|.$$ (10)

Therefore, the ride comfort evaluation index $I_c$ can be defined as:

$$I_c = \sum_{i=1}^{n} \begin{cases} 0 & \Delta u_i \leq \Delta u^E \\ \Delta u_i & \Delta u_i > \Delta u^E \end{cases}.$$ (11)

where $\Delta u^E$ is the threshold for acceleration change.

- *Parking accuracy*: It is used to assess the parking accuracy, expressed as:

$$e_s = |s_i - s_D|.$$ (12)

where $s_D$ is the length of the segment between adjacent stations and $s_i$ is the current running distance of the train. Note that the parking error of the metro is generally required to be within $\pm 30$ cm [21] so that metro barrier doors can be opened. Therefore the parking accuracy index can be defined as:

$$I_p = \begin{cases} 1 & e_s \leq 30, v_i = 0 \\ 0 & e_s > 30, v_i = 0 \end{cases}.$$ (13)

## 2.2 Problem objectives

This section will clearly articulate the problems that this study aims to address. The two proposed EITO algorithms (EITO$_E$ and EITO$_P$) aim to achieve the following objectives corresponding to the above - mentioned problems:

1. **Meeting multi-objective requirements:** The EITO algorithms should be able to provide control strategies for traction and braking forces that can meet the requirements of multiple objectives such as safety, comfort, punctuality, parking accuracy, and energy efficiency of metro operation. Given the definitions of safety ($I_s$), punctuality ($I_t$), energy efficiency ($I_e$), comfort ($I_c$), and parking accuracy ($I_p$), the algorithms need to ensure that the train operation satisfies all these evaluation indices simultaneously.

2. **Independent of offline speed profile and continuous force control:** The EITO algorithms should be able to perform normal operations without considering the speed distribution of the offline design and achieve the control of continuous forces. As existing ATO systems mainly rely on offline-designed speed profiles and current intelligent driving algorithms have limitations in continuous force control, the EITO algorithms aim to overcome these drawbacks.

3. **Outperforming existing methods in energy-efficiency and comfort:** The control strategy output by the EITO algorithms should outperform experienced metro drivers and current intelligent driving algorithms in terms of energy efficiency while ensuring good ride comfort. By comparing with manual driving and existing intelligent driving algorithms (such as ITOR and STON), the EITO algorithms should achieve lower energy consumption ($I_e$) and better comfort ($I_c$) performance.

4. **Adapting to different situations:** The EITO algorithms should be able to flexibly adapt to different situations, including different trip times, different temporary faults (earlier

or later arrival), speed limits, and gradients conditions (simple or complex). Considering the complex and variable operating conditions of metro trains, the algorithms need to adjust their control strategies accordingly to ensure stable and efficient operation.

Existing ATO systems must track the designed offline speed profile, and current intelligent driving algorithms either cannot achieve control of continuous forces or cannot adapt to complex and variable line conditions, which is the driving force behind this paper. Moreover, RL has been applied in many fields to deal with model-free problems [24], and expert knowledge has been widely used to improve control strategies [10],[11]. Therefore, in this paper, two intelligent algorithms are proposed. Namely, $EITO_E$ and $EITO_P$, where $EITO_E$ is a heuristic algorithm based on an expert system to address multiple performance objectives of metro train operation. In addition, we develop $EITO_P$ based on $EITO_E$ using the PPO to comprehensively optimize the multi-objective requirements of safety, comfort, punctuality, parking accuracy, and energy efficiency.

Following the problem statement outlined above, the next section will provide a detailed introduction to the specific control models and methodologies employed to achieve these objectives.

## 3 EITO algorithm design

The application of expert experience-based control methods to automatic train operation control is motivated by the following two reasons. On the one hand, because the train operation control system is a highly complex, multiobjective nonlinear dynamical system [23,25], which poses great difficulties for traditional control that requires the use of its precise mathematical model; on the other hand, experienced drivers combined with their long-term accumulated maneuvering experience can implement effective control of the train in real-time so that the train operation meets the requirements of several control objectives [26].

Therefore, we first developed an expert system-based $EITO_E$ algorithm. This expert system contains expert rules and a heuristic inference system. These expert rules were summarized by our communication with metro drivers and by analyzing data from YLBS and literatures. In addition, we developed a heuristic inference method to solve without an offline speed profile reference based on the driver's operating strategy. Then, the appropriate $EITO_E$ output is obtained by combining the speed limit and the current state of the train.

Both $EITO_E$ and $EITO_P$ ensure punctuality through real-time adjustments based on current train conditions and speed limits. $EITO_E$ uses expert rules to allocate trip times effectively, while $EITO_P$ employs reinforcement learning to dynamically adjust acceleration and braking strategies. The algorithms continuously monitor the train's position and speed, allowing them to make timely decisions that keep the train on schedule. Specifically, the reward function in $EITO_P$ penalizes deviations from planned trip times, reinforcing behaviors that promote punctuality.

### 3.1 $EITO_E$

This research adhered to strict ethical standards. In data collection (human or animal), we followed relevant regulations and obtained necessary consents. Experienced drivers can meet multiple objectives well. By observing the driver's behavior, we found that an experienced driver can control the train in the correct position, allocate the reserved time reasonably, avoid unnecessary braking, limit the train speed to prevent over-speeding, and reduce the

number of switches in the controller output. Based on the study of [10],[12], we derived IF-THEN rules using position, speed, and running time as inputs and acceleration/braking rate as outputs. These rules can be described as follows.

I. Energy-efficient trains operate in three states, namely acceleration, coasting, and braking. The train does not transition directly from the acceleration state to the braking state and vice versa, unless a special incident is encountered. Transfer between any other two states is allowed.

II. The acceleration of the train starting process should be appropriate for comfort (usually less than 0.6 m/s$^2$ ).

III. For better comfort, the rate of change of acceleration in each time interval should not be too large (usually less than 0.3 m/s$^3$ ).

IV. Determine the next operation mode in advance according to the current speed and the next speed limit value to avoid triggering automatic train protection.

V. Allocate the total trip time to each interval according to the speed limit, and try to operate according to the allocated time in each interval.

As mentioned before, experienced drivers consider the train's reserved time, reserved distance, speed limit, and current speed: if the train's speed is too low to arrive in time, the train will accelerate. Conversely, if the train's speed is too high, the train coasts. We designed a data-driven inference method DMTD to determine the coasting or accelerating time. This inference method is manually driven and uses the twice MTD to calculate the desired speed range ($v_{ik}^{\mathrm{desired}}$ ) for the current speed limit interval. As shown in Fig 3, the $v_{ik}^{\mathrm{desired}}$ calculated by this method enables energy-efficient driving by making the train coast as much as possible while ensuring punctuality.

Using the online data of the train, we first use the DMTD algorithm (see Algorithm 1) to obtain the appropriately reserved trip times $t_1^r$ and $t_2^r$ in the current speed limit interval. Then,

**Algorithm 1. DMTD algorithm.**

1: Get online and offline data including current train location $s_i$, speed $v_i$ (point in Fig 2) and speed limit $V_i^{\mathrm{limit}}$ . Reserve travel time $T_{r1} = T - t_i - T_0, T_{r2} = T - t_i + T_0$, assuming $s_{k-1}^{\mathrm{limit}} \leq s_i \leq s_k^{\mathrm{limit}}$ , which means that the train is already in the speed limit interval $\left(s_{k-1}^{\mathrm{limit}}, s_k^{\mathrm{limit}}\right)$ .

2: Every time the train enters a speed limit interval, as shown in the red dot of Fig 2 (indicating that the train enters the second speed limit interval), we draw the maximum traction speed curve from the train position and each speed limit section. Then, from the left end of each speed limit segment, the maximum brake speed curve is drawn to obtain the minimum travel time curve.

3: Calculate the minimum reserved time $t_r^{\mathrm{min}}, T_r^{\mathrm{min}}$ from the minimum travel time curve between the current position $S_i$ and leaving the current speed limit interval and reaching the destination.

4: Calculate the reserve time for the current speed limit interval $t_1^r = T_{r1} \frac{t_r^{\mathrm{min}}}{T_r^{\mathrm{min}}}, \quad t_2^r = T_{r2} \frac{t_r^{\mathrm{min}}}{T_r^{\mathrm{min}}}$ .

5: **Return** $t_1^r, t_2^r$ .

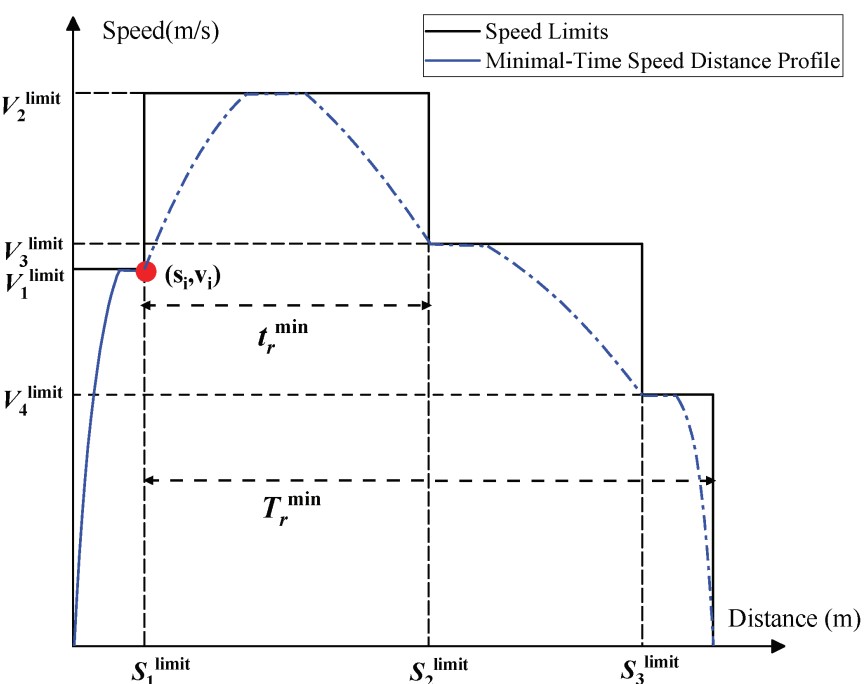

**Fig 2. DMTD algorithm.**

the estimated velocity range for the speed limit of each segment is calculated from the formula below $\left[ v_{i1}^{\text{desired}}, v_{i2}^{\text{desired}} \right]$.

$$v_{ik}^{\text{desired}} = \frac{s_k^{\text{limit}} - s_i}{t_k^r}. \tag{14}$$

The DMTD algorithm and Eq (14) indicate that if the train is between $\left[ v_{i1}^{\text{desired}}, v_{i2}^{\text{desired}} \right]$, the train can reach its destination on time (within ±3 s of the planned trip time is on schedule, setting $T_0$=3 in the Algorithm 1). Therefore, if the train's speed is lower than $v_{i1}^{\text{desired}}$, the train needs to accelerate; if the train's speed is higher than $v_{i2}^{\text{desired}}$, the train should coast. Then, the reasoning for determining the mode of operation (coasting or accelerating) is summarized as follows.

1) If $v_i < v_{i2}^{\text{desired}}$, the train should accelerate, and the output of the expert system is defined as Eq (15):

$$u_{i+1}^E = \begin{cases} u_i^E + \Delta u_i^E & u_i^E < u_{\max} \\ u_{\max} & u_i^E \leq u_{\max} \end{cases}. \tag{15}$$

where $u_{\max}$ is the maximum acceleration, and $\Delta u_i^E$ is the rate of variation of the acceleration in the time interval $\Delta t$. Note that the parameter $\Delta u_i^E$ is setted as 0.3 m/s$^3$ according to expert experience. If unknown disturbances are considered, such as the resistance and gradient of the line, $\Delta u_i^E$ is not constant. And the value of this parameter will be adjusted by PPO in the next section.

2) If $v_i \geq v_{i2}^{\text{desired}}$, the train should coast and the output of the expert system can be described as Eq (16):

$$u_{i+1}^E = 0. \tag{16}$$

In addition, when the speed limit $v_{j+1}^{\text{limit}}$ of the next section is less than the speed limit $v_j^{\text{limit}}$ of the current section, as shown in Fig 3, the train may need to brake at a reasonable speed to ensure the safety of the train. In other words, the speed of the train should always be lower than the speed limit. In this case, we define the safe speed $v_i^{\text{safe}}$ to monitor the speed of the train:

$$v_i^{\text{safe}} = \sqrt{\beta \left( V_{j+1}^{\text{limit}} \right)^2 - 2u_{\min} \left( s_j^{\text{limit}} - s_i \right)} \tag{17}$$

where $s_i$ is the current position of the train. $s_j^{\text{limit}}$ is the starting position of the next section. $\beta$ is the speed scaling factor caused by the time delay and friction of the railroad, which is taken as 0.95 in this paper. $u_{\min}$ is the maximum deceleration speed. In this paper, $u_{\min} = -1$ m/s$^2$. When the train operates to the position indicated by the mark 1 in Fig 3, i.e., when the current speed $v_i$ is higher than or equal to $v_i^{\text{safe}}$, then the train should immediately apply the maximum deceleration $u_{\min}$. In addition, if the length of the current speed limit interval is long enough, there will be a situation where $v_i^{\text{safe}}$ exceeds the speed limit, which may cause the train to run beyond the speed limit (shown in Fig 3, marker 2.), so we redefine the safe speed to ensure safe driving:

$$v_i^{\text{safe}} = \min \left\{ \sqrt{\beta \left( V_{j+1}^{\text{limit}} \right)^2 - 2u_{\min} \left( s_j^{\text{limit}} - s_i \right)}, \beta V_j^{\text{limit}} \right\}. \tag{18}$$

We defined the parking accuracy (parking error less than ±30 cm) in Eq (13), and all three automatic stop control algorithms (TASC) proposed in our previous work can achieve accurate parking of trains, the details of TASC please see the reference [25]. Therefore, we apply the heuristic online learning algorithm (HOA) of TASC at the location shown in Fig 3 mark

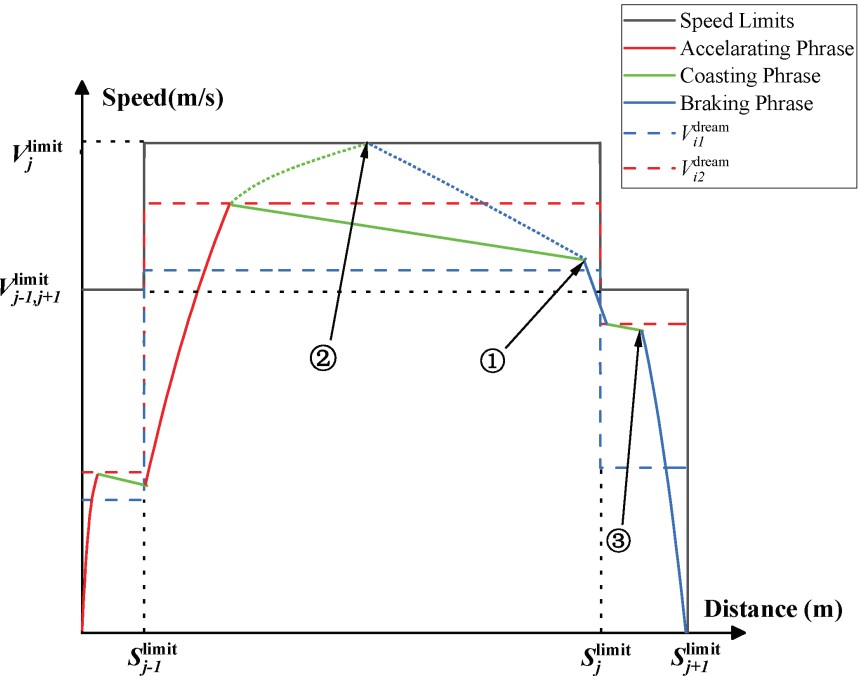

**Fig 3. EITO$_{\text{E}}$ speed distance profile.**

3 to ensure the parking accuracy of the train. The EITO's expert system is implemented after expert rules and heuristic inference methods have been designed. As illustrated in Fig 3, EITO$_\text{E}$ can make appropriate acceleration, coasting, or deceleration decisions based on online and offline data such as speed limits and gradients, as well as expert reasoning methods, and its speed profile can be divided into acceleration phase, multiple coasting phases, safety braking phase, and parking phase. Furthermore, the output is constrained by expert criteria to assure comfort and punctuality.

However, EITO$_\text{E}$ cannot optimize energy consumption online, because $\Delta u_i^E$ is specified as a constant value. Therefore, the PPO is introduced to improve the performance of EITO$_\text{E}$.

## 3.2 EITO$_\text{P}$

RL is a machine learning paradigm that aims to learn to control systems in environments to maximize numerical performance associated with long-term goals [27]. Three reasons motivate us to adapt deep reinforcement learning in train control tasks:

(1) The EITO algorithm does not require reference to the target speed profile, while RL does not require external supervision. (2) During train control, behavior affects not only the immediate reward but also the reward for future states, which falls into the advantage of RL. (3) The use of deep reinforcement learning can modify the control strategies used in current ATO systems for discrete actions. *EITO$_P$*. The algorithmic process of EITO$_P$ is presented in Algorithm 2.

Markov Decision Process: Before applying the reinforcement learning algorithm, we formulate our problem as a Markov Decision Process (MDP), which provides a mathematical framework for decision making. The key elements of reinforcement learning include its state, action, policy, and reward, which are defined as follows.

- *State $x_i$.* In this case, the train status, with the current position, speed, and reserved trip time, can be described as:

$$x_i = \left[ s_i, v_i, T_p - t_i \right], \ i = 0, 1, 2, \cdots, m. \tag{19}$$

Let $x_0$ denotes the initial train state and $x_m$ denotes the final state. Obviously, the following equations should hold:

$$x_0 = \left[ 0, 0, T_p \right]. \tag{20}$$

$$x_m = \left[ S, 0, T_p - t_m \right] \tag{21}$$

- *Action $a_i$.* In contrast to $\Delta u_i^E$ in Eq (15) in EITO$_\text{E}$, EITO$_\text{P}$ has a variable variation of acceleration in each state. As shown in Eqs (22)–(23), we use $\Delta u_i^\text{var}$ instead of $\Delta u_i^E$. Meanwhile, we define the range of $u_i$ and $\Delta u_i^\text{var}$ as [–1,1] and [–0.3,0.3], respectively. Therefore, action $a_i$ can be defined for the EITO$_\text{P}$:

$$a_i = \Delta u_i^\text{var}. \tag{22}$$

when $v_i < v_{i2}^\text{desired}$ :

$$u_i = \begin{cases} u_{i-1} + \left| \Delta u_i^\text{var} \right| & u_{i-1} < u_\text{max} \\ u_\text{max} & u_{i-1} \geq u_\text{max} \end{cases} . \tag{23}$$

when $v_i \geq v_{i2}^\text{desired}$ :

$$u_i = \Delta u_i^\text{var}. \tag{24}$$

**Algorithm 2. Detailed process of EITO$_P$.**

```
Initialize reply buffer capacity T;
Randomly initialize Actor-New, Actor-Old networks and
assign random weights θ,θ′, and θ′=θ;
Randomly initialize the Critic network and assign a random
weight φ.
```
**for** $i=1,\cdots,N$ **do**
 **for** $j=0,\cdots,T$ **do**

 The environment information $x_j$ is input to Actor-New
network, and then an action $a_j$ is sampled out by a normal
distribution (the obtained action $a_j$ is verified using expert
knowledge and inference methods.

 If the action does not meet these requirements, the
obtained action $a_j$ is adjusted according to the metro
equation of motion.

 Then $a_j$ is executed.

 The result is then input to the environment to obtain
the reward $r_i$ and the next state $x_{j+1}$.

 The tuple $\left(x_j,a_j,r_j\right)$ is then stored in the reply buffer
$\left[\left(x_j,a_j,r_j\right),\cdots\right]$.

 Finally, $x_{j+1}$ is input to the Actor-New network.

 **end for**

 Input $x_t$ into the critic network to obtain the advantage
function $V_\phi(x_t)$, $A_t=\sum_{t>t'}\gamma^{t'-t},r_{t'}-V_\phi(x_t)$

 Update the Critic network parameters by minimizing the
loss function:
$$L_c=\frac{1}{N}\left(\sum_{t>t'}\gamma^{t'-t}r_{t'}-V_\phi(x_t)\right)^2$$
 **for** $k=0,\cdots,M$ **do**

 The combination of all states $x_i$ stored in the buffer
is input into the Actor-New, Actor-Old network to obtain
two state action probability rate distribution $\pi_\theta(a_k|x_k)$,
$\pi_{\theta'}(a_k|x_k)$, and calculate: $r_k(\theta)=\frac{\pi_\theta(a_k|x_k)}{\pi_{\theta'}(a_k|x_k)}$

 Update the actor-new network weights $\theta$ by Eq (31).

 **end for**

 Update the actor-old network with the actor-new network
weights: $\theta'=\theta$.

**end for**

- *Policy $\pi$*. The policy $\pi$ represents the probability of taking an action while processing a discrete action task. In this paper, since EITO is intended to address continuous action control tasks, a policy is a statistic of the probability distribution, which is expressed as Eq (25):

$$\pi(a\,|\,x,\theta)=\mathcal{N}(\mu(x,\theta),\sigma(x,\theta)) \tag{25}$$

where $\theta$ is the weight.

- *Reward function $r(x_i,a_i)$*: This function defines the reward that the train receives when it takes an action in a given state. In this case, our reward function is defined by the time error

$\Delta t_e$, the passenger comfort $\Delta I_c$ and the energy consumed per unit mass $\Delta I_e$ in the time interval when the train takes action $a_i$ in state $x_i$.

$$r(x_i, a_i) = -k_1 \Delta I_e - k_2 \Delta I_c - t_e \Delta t_e \tag{26}$$

where $k_1, k_2$ are determined by expert experience, and $t_e$ is defined as Eq (27):

$$t_e = \begin{cases} 0 & t_i \le T_p \\ 1 & t_i > T_p \end{cases} \tag{27}$$

The role of $\Delta t_e$ and $\Delta I_c$ is used to ensure that the agent optimizes energy consumption while ensuring punctuality and comfort as much as possible, rather than just reducing energy consumption.

The EITO$_P$ algorithm is based on the PPO algorithm [28]. PPO is a deep reinforcement learning algorithm based on policy gradient (PG). Moreover, it is based on the Actor-Critic framework capable of handling continuous action control and model-free problems. The PPO algorithm limits the update magnitude of the new policy according to the ratio of the old to the new policy so that the PG algorithm can be trained and converge at a larger learning rate. The objective function of the policy gradient algorithm is:

$$J(\theta) = E_i \left[ A(x_i, a_i) \pi_\theta (a_i \mid x_i) \right]. \tag{28}$$

where $\pi(\bullet)$ denotes the policy function; $\theta$ is the network parameter of Actor; $i$ denotes the state or action of the $i$th step; $A(x_i, a_i)$ is the estimate of the advantage function of $i$th step, as shown in Eq (29); $E$ denotes the empirical expectation of the time step. The advantage function is chosen at the state to compare the obtained score with the average score. If it is high, then the advantage function is positive. Otherwise, it is inverse. The gradient ascent method is used to update the value function.

$$A(x_i, a_i) = Q_\pi(x_i, a_i) - V_\pi(x_i). \tag{29}$$

where $Q_\pi(x_i, a_i)$ is the state action-value function, which represents the expected reward of the Agent following the policy $\pi$, after performing an action $a_i$ in state $x_i$ until the end of the episode. Similarly, the state value function $V_\pi(x_i)$ represents the expected reward of the Agent following the policy from the state $x_i$ to the end of the episode.

Because the PG algorithm adopts the online update policy to resampling every parameter update, its learning rate is not easy to determine. The PPO algorithm converts the online update strategy into an offline update strategy, i.e., a new and old Actor strategy is used. The training data of the new Actor can be obtained from the old Actor, while the new strategy weight is expressed using the ratio of action probabilities $r_i(\theta)$ of the old and new strategies, which is expressed as Eq (30):

$$r_i(\theta) = \frac{\pi_\theta(a_i \mid x_i)}{\pi_{\theta'}(a_i \mid x_i)}. \tag{30}$$

where $\theta'$ is the sampled neural network parameter. If the probability distributions obtained for two neural network parameters $\theta$ and $\theta'$ in the same state differ greatly and in the case of under-sampling, it leads to a large variance between them. Therefore, the PPO algorithm adds a CLIP function to the base of the objective function to limit the parameters $\theta$ and $\theta'$, given as

follows:

$$J_{PPO} = E_{(x_i,a_i) \sim \pi_\theta} \left[ \min \left( r_i(\theta) A_t, \ \text{clip} \left( r_i(\theta), |1 - \varepsilon| \right) A_t \right) \right]. \tag{31}$$

The term "continuous control task" in our work refers to real - time optimization of continuous traction/braking forces without relying on predefined discrete actions or offline speed profiles. Unlike traditional methods that track fixed trajectories, our algorithms dynamically adjust acceleration/braking rates (Eq (22-(24)) based on real - time states (position, speed, remaining time) and environmental conditions (speed limits, gradients). This is enabled by:

- $EITO_E$: Expert rules (Sect 3.1) and the DMTD heuristic (Algorithm 1) generate smooth, continuous force adjustments.
- $EITO_P$: The PPO - based reinforcement learning framework (Sect 3.2) optimizes continuous actions ($\Delta u_i^{\text{var}}$) in a policy gradient manner (Eq (25), allowing fine - grained control over acceleration/deceleration.

This approach eliminates abrupt state transitions (e.g., discrete coasting points in prior works [10, 12]) and ensures seamless adaptation to varying line conditions (Sect 4.3).

While the core control logic is detailed in Algorithms 1 ($EITO_E$) and 2 ($EITO_P$), we acknowledge that the convergence analysis of PPO training could be elaborated further. For clarity: Control Steps:$EITO_P$ iteratively samples actions from a Gaussian policy (Eq (25) and updates actor-critic networks using clipped surrogate objectives (Eq (31). The reward function (Eq (26) penalizes energy consumption, comfort violations, and time deviations, ensuring balanced optimization. Convergence: Fig 5 (training curves) shows energy consumption and running time stabilize after 80 episodes, indicating policy convergence.

## 4 Simulations

To verify the intelligence, flexibility, and robustness of $EITO_E$ and $EITO_P$, we designed three numerical simulation experiments based on field data collected in YLBS. YLBS started operation in Beijing on December 30,2010, with a total length of 23.3 km, starting from Songjiazhuang station and ending at Ciqu station. The train type used in YLBS is DKZ32 EMU with 6 vehicles, whose parameters are shown in Table 1. To rigorously validate the suitability of $EITO_P$ for online control, we conducted experiments on a workstation with the following specifications: CPU: Intel i9-10900K (10 cores, 3.7 GHz) , GPU: NVIDIA RTX 3090 (24 GB VRAM), Memory: 64 GB DDR4 , Software: Python 3.8, TensorFlow 2.6.

Three simulation cases are presented in this section. The manual driving dataset we use in this section was collected in YLBS from May 1, 2015, to May 27, 2015, including 100 groups of up trains and down trains. We select the manual driving data with the best-generalized performance from the recorded dataset as $EITO_M$. In Case 1, we compare the results of all algorithms ($EITO_M$, ITOR, STON, $EITO_E$ and $EITO_P$). In Case 2, we test the intelligence and flexibility of all algorithms by varying the planned trip time of the same rail segment. In Case 3, we test the operational performance of EITO models with complex gradients and speed limits to verify the robustness of proposed $EITO_E$ and $EITO_P$.

### 4.1 Case 1

Taking the interval between Rongjing station(RJ) and Wanyuanjie station(WYJ) as an example, the speed limit and gradient of this interval are shown in Fig 4. The planned trip time $T_p$ =101 s is the same as the actual operation, and the distance between the two stations is 1280 m.

**Table 1. Parameters of DKZ32.**

| Parameters | Value |
|---|---|
| $M(\ kg)$ | $1.99 \times 10^5$ |
| $m_i, i = 1, 6(\ kg)$ | $3.3 \times 10^4$ |
| $m_i, i = 3(\ kg)$ | $2.8 \times 10^4$ |
| $m_i, i = 2, 4, 5(\ kg)$ | $3.5 \times 10^4$ |
| $\Delta \ddot{l}_i, i = 1, 3, 6(\ mm)$ | $0.1 \sin(t)$ |
| $\Delta \ddot{l}_i, i = 2, 4, 5(\ mm)$ | $0.15 \cos(t)$ |
| Time constant (Braking) $\sigma$ | 0.4 |
| Time delay (Braking)(s) $\tau$ | 0.8 |
| Time constant (Accelerating) $\sigma$ | 0.4 |
| Time delay (Accelerating)(s) $\tau$ | 1 |
| $\alpha$ | 1.244 |
| $\beta, \gamma$ | $\left(1.45 \times 10^{-2}, 1.36 \times 10^{-4}\right)$ |

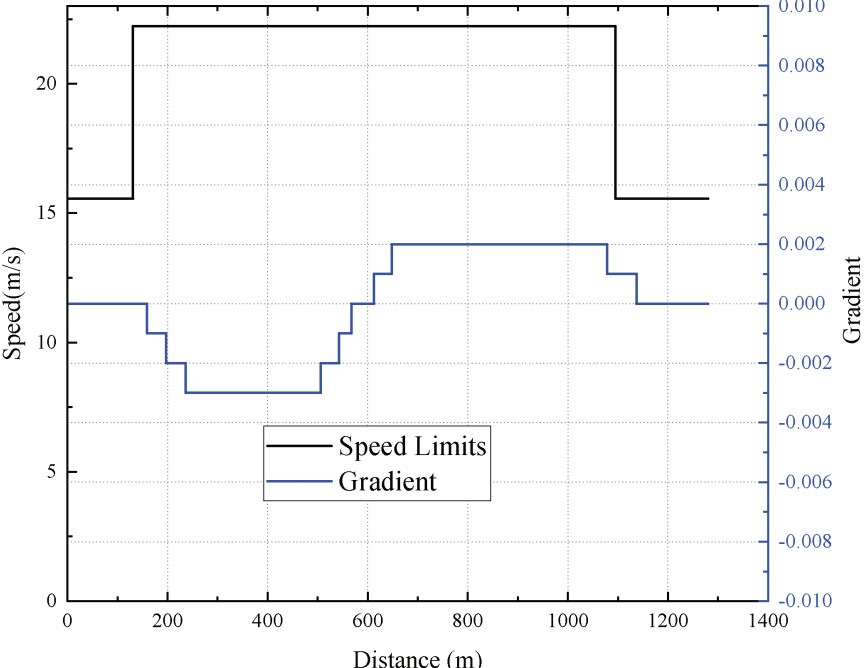

**Fig 4. Speed limits and gradients from RJ to WYJ.**

Figs 5 and 6 show the energy consumption *E* and the running time (/s) during the online learning process of EITO$_P$. The results show that during DRL, the energy consumption is reduced from 380 to 364 after about 80 rounds of training and gradually approaches the optimal value. In addition, the running time floats within 100 s~102 s ($T_p$ =101 s). According to the definition of punctuality i.e., a time error of less than $\pm 3$ s is allowed. This indicates that applying the PPO algorithm in EITO$_E$ can reduce the energy consumption online while satisfying the running time error constraint.

It can be seen from Fig 7 that the EITO$_E$ and EITO$_P$ start to coast after accelerating to $v_{i2}^{\text{desired}}$ in the first two speed limit intervals, and the coast point of EITO$_P$ is more advanced. The coast distance of EITO$_E$ and EITO$_P$ are 899.78 m and 842.00 m, respectively. Note that in the last speed limit interval, EITO$_P$ did not choose to coast at the position where EITO$_E$

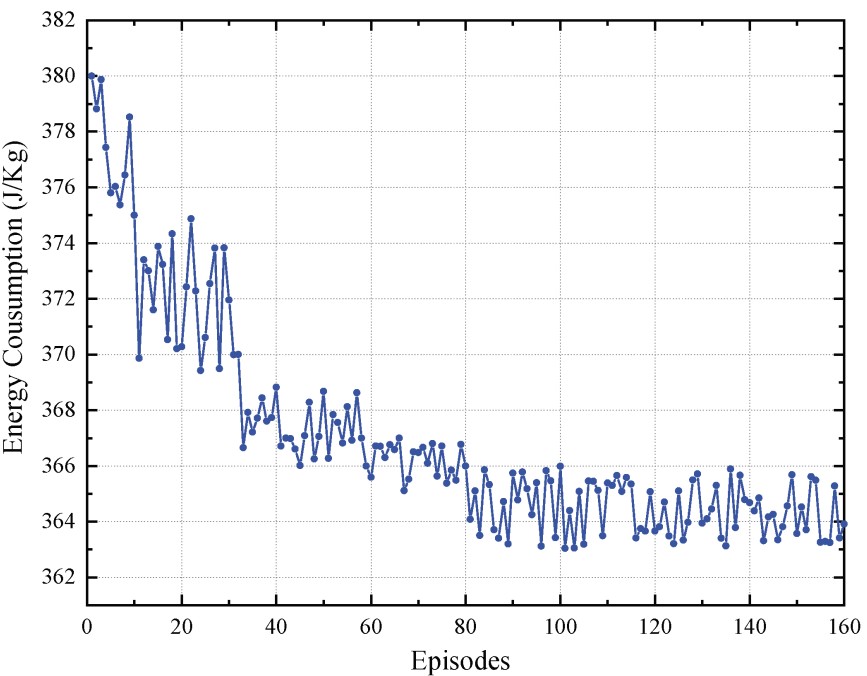

**Fig 5. Energy consumption in the PPO process.**

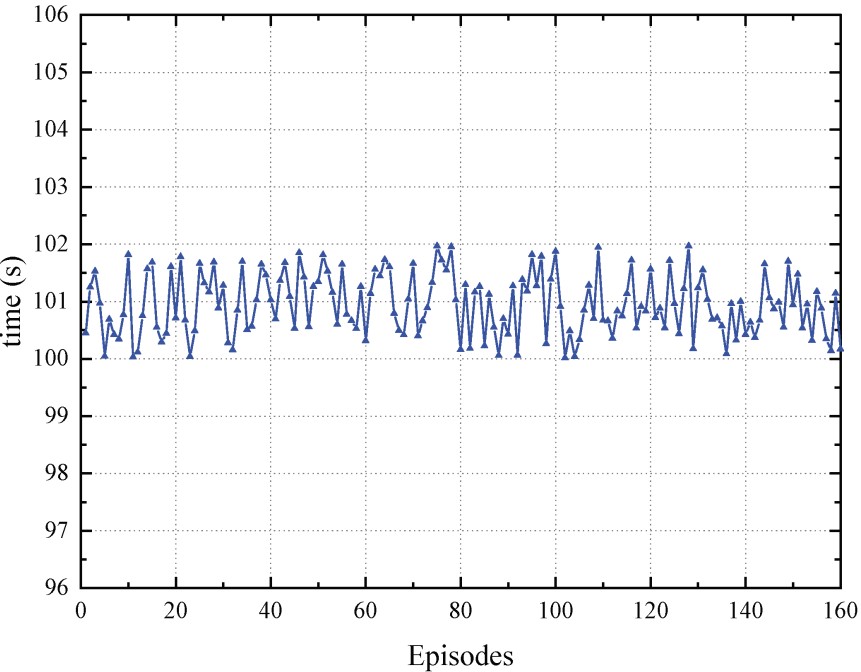

**Fig 6. Running time in the PPO process.**

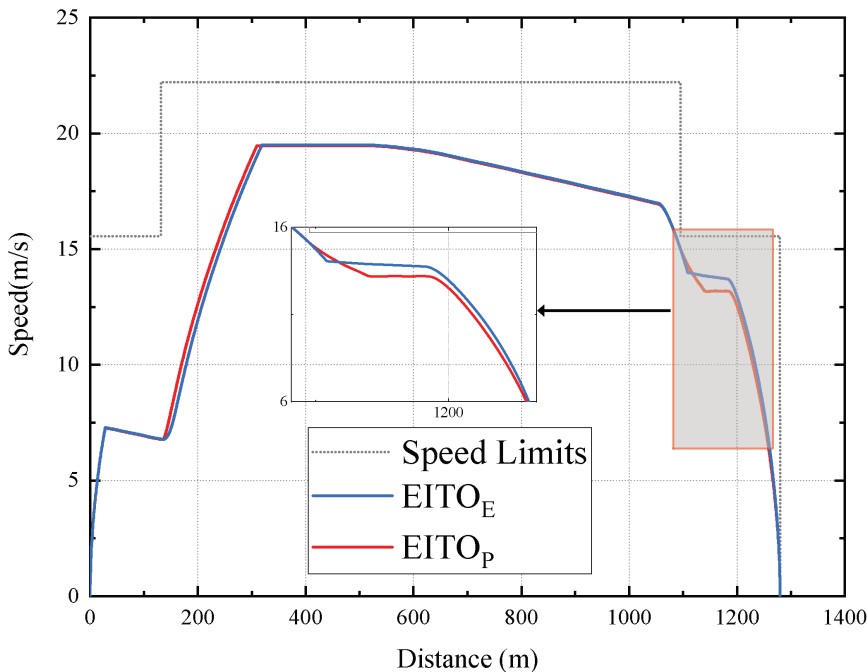

**Fig 7. Operation curve comparison among the EITO algorithms.**

started to coast. Instead, it decelerates slightly based on its current position, speed, and remaining trip time. It causes EITO$_P$'s $I_e$ and $I_c$ can be further reduced compared to EITO$_E$, which shows that EITO$_P$ can consider the constraints of mult-objectives in a more integrated way.

Figs 7 and 8 shows the speed distance curves of the five algorithms at a 101 s planned trip time. It can be seen the speed curve of EITO$_M$ can be divided into the full acceleration phase, coasting phase, and full braking phase. EITO$_E$ has the highest maximum speed of 19.49 m/s, with four phases, the acceleration phase, multiple coasting phases, safety braking phase, and parking phase in its speed-distance curve. In addition, the coasting distances of EITO$_M$, ITOR, and STON are 677.99 m, 398.78 m, and 661.46 m respectively, which are significantly less than those of EITO$_E$ and EITO$_P$, indicating that the proposed EITO algorithms have lower energy consumption.

We can see from Table 2 that all five algorithms meet the requirements of YLBS in terms of safety, punctuality, and parking accuracy. Among these algorithms, ITOR has the highest energy consumption. Compared with EITO$_M$, ITOR is 1.7% higher than EITO$_E$; STON is 11.7% lower than EITO$_M$, EITO$_E$ is 34.9% lower than EITO$_M$, and EITO$_P$ can further optimize energy consumption by 4.3% based on EITO$_E$. In terms of comfort for all algorithms, EITO$_M$ has the highest $I_c$, indicating the worst passenger comfort, while the rest of algorithms have similar values for the comfort index, which is much less than EITO$_M$. And EITO$_P$ has the best $I_e$ and $I_c$ in 101 s trip time.

## 4.2 Case 2

In this case, we verified the flexibility of five algorithms in 95 s planned trip time and 115 s planned trip time by simulating different planned trip times in the same railroad section.

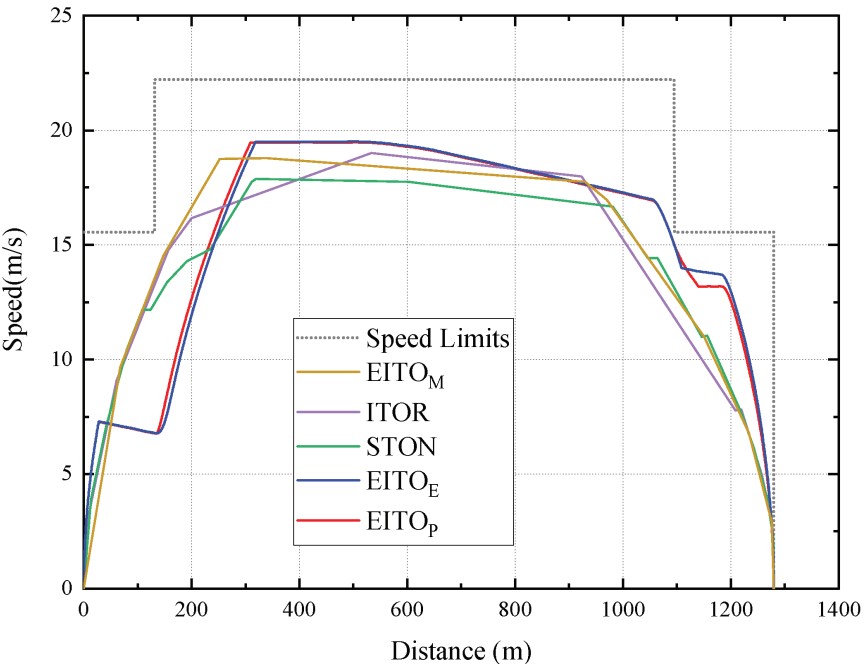

**Fig 8. Operation curve comparison among the five algorithms in the 101s trip.**

**Table 2. Comparison of performance with different trip time.**

| Performance | t | $I_t$ | $I_s$ | $I_e$ | $I_c$ | $I_p$ |
|---|---|---|---|---|---|---|
| $EITO_M$(101 s) | 102 s | 1 | 1 | 583.82 | 9.21 | 1 |
| ITOR(101 s) | 102 s | 1 | 1 | 597.21 | 4.55 | 1 |
| STON(101 s) | 102 s | 1 | 1 | 518.03 | 4.32 | 1 |
| $EITO_E$(101 s) | 101 s | 1 | 1 | 379.52 | 3.05 | 1 |
| $EITO_P$(101 s) | 101 s | 1 | 1 | 362.37 | 2.40 | 1 |
| $EITO_M$(95 s) | 96 s | 1 | 1 | 811.77 | 14.00 | 1 |
| ITOR(95 s) | 97 s | 1 | 1 | 854.21 | 8.8 | 1 |
| STON(95 s) | 96 s | 1 | 1 | 740.29 | 5.27 | 1 |
| $EITO_E$(95 s) | 97 s | 1 | 1 | 436.51 | 2.09 | 1 |
| $EITO_P$(95 s) | 96 s | 1 | 1 | 438.31 | 2.92 | 1 |
| $EITO_M$(115 s) | 116 s | 1 | 1 | 325.05 | 7.50 | 1 |
| ITOR(115 s) | 116 s | 1 | 1 | 326.58 | 3.80 | 1 |
| STON(115 s) | 115 s | 1 | 1 | 620.06 | 4.01 | 1 |
| $EITO_E$(115 s) | 115 s | 1 | 1 | 340.64 | 4.03 | 1 |
| $EITO_P$(115 s) | 114 s | 1 | 1 | 316.42 | 2.95 | 1 |

Since an ATO system generally needs to have offline speed recommendation curves, it is difficult to dynamically adjust the trip time. Furthermore, increasing regenerative energy requires real-time reprogramming of the planned trip time for each train on the metro line. If the train model can adjust the arrival time in real-time according to the notification, the regenerative energy can be better utilized to achieve the energy-saving operation of the metro [29],[30].

Therefore, we similarly carried out two examples of dynamically adjusting the trip time (extending or reducing the trip time) by using $EITO_P$ to overcome the above shortcomings. In our simulations, such examples are called EITO with flexible adjustable trip times.

Fig 9 shows the speed distance curves for the five algorithms at a trip time of 95 s. ITOR has the highest maximum velocity of 22.22 m/s and a shorter coasting distance, indicating that ITOR may have higher energy consumption and worse passenger comfort. STON has the lowest maximum velocity, but it decelerates too early in the second speed limit interval, resulting in a shorter coasting distance and higher energy cost. The speed distribution curves of $EITO_E$ and $EITO_P$ are similar as they are both smoother and have a longer coasting distance, indicating that both algorithms may perform better in terms of comfort and energy consumption. In addition, $EITO_P$ accelerates slightly in the last section where the speed-limited $EITO_E$ coasts, indicating that $EITO_P$ can adjust the arrival time of $ETTO_E$, which further illustrates the effectiveness of $EITO_P$.

It can be seen from Table 2 that compared with $EITO_M$, the energy consumption of ITOR is 5.2% higher than that of $EITO_M$; the energy cost of STON is 8.8% lower than that of $EITO_M$; the EITO algorithms perform more superiorly, both saving more than 45% in energy cost compared with $EITO_M$. In addition, in terms of riding comfort of the five methods, $EITO_M$ has the largest $I_c$, while ITOR, STON, and $EITO_E$ have similar $I_c$, which is much smaller than the $EITO_M$. $EITO_P$ has the best comfort with 2.92.

Fig 10 shows the speed distance curves for the five algorithms at a trip time of 115 s. The maximum speed of $EITO_M$ is 14.73 m/s, the maximum speed of ITOR and STON are 14.69 m/s and 14.65 m/s, respectively. The maximum speed of $EITO_E$ and $EITO_P$ are 16.15 m/s and 16.10 m/s, respectively. We can see that, compared with the speed distance curves at 95 s and 101 s planned trip times, their maximum velocities are much lower than the previous cases, indicating that they have lower average velocities and lower energy consumption.

Furthermore, we can learn from Table 2 that all five algorithms meet the requirements in terms of safety, punctuality, and parking accuracy. Compared with $EITO_M$, the energy

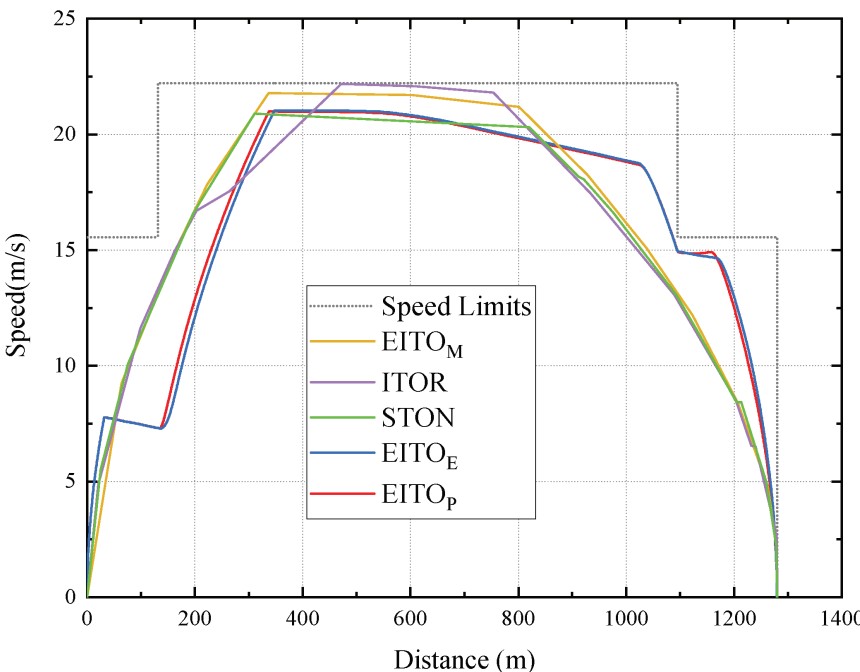

**Fig 9. Operation curve comparison among the five algorithms in the 95 s trip.**

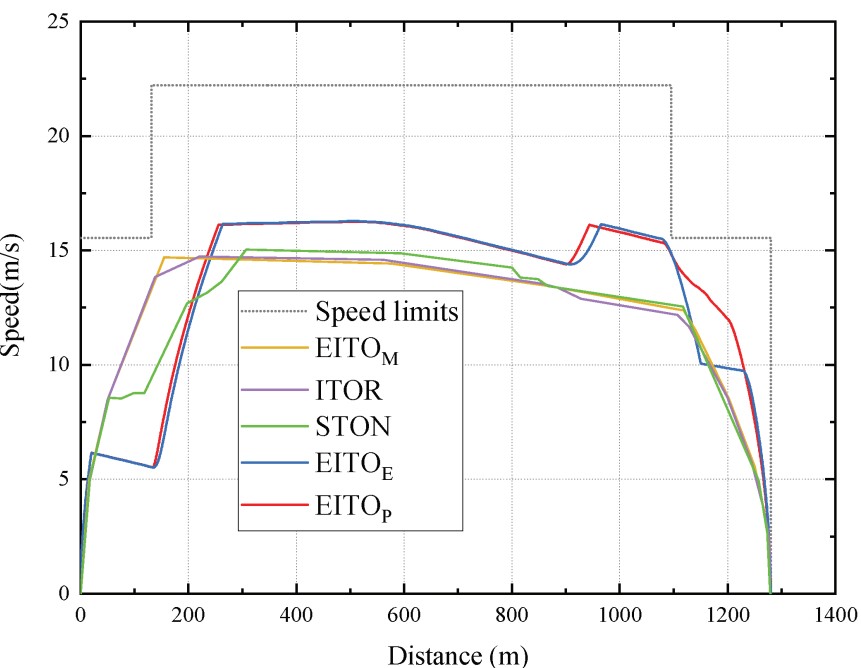

**Fig 10. Operation curve comparison among the five algorithms in the 115 s trip.**

consumption of ITOR is 0.5% higher than $EITO_M$, and the energy consumption of $EITO_E$ is 4.8% higher than $EITO_M$. The energy consumption of STON is 1.5% lower than $EITO_M$, and the energy consumption of $EITO_P$ is 2.7% lower than $EITO_M$. It is easy to see that $EITO_E$ has the highest energy consumption. Meanwhile, the $EITO_E$'s comfort is also higher than the other three Intelligent operation models due to its multiple and large changes in $u$ during the acceleration phase. In addition, although $EITO_P$ arrived 1 s earlier than the expected arrival time, the $I_c$ and $I_e$ of $EITO_P$ are better than $EITO_E$. This result further illustrates that $EITO_P$ can dynamically optimize the train's operating state, comprehensively considering the constraints of multiple objectives. In this instance, $EITO_P$ outperforms the other four algorithms in $I_c$ and $I_e$.

Fig 11 shows the running curve of the $EITO_P$ with dynamically adjusted (earlier or later) arrival time on the RJ to WYJ rail section and the originally planned trip time is 101 s. It should be noted that 15 s Later is the speed curve where the train is informed of the 15 s later arrival and 10 s Earlier is the speed curve where the train is informed of the 10 s earlier arrival. The Constant trip is the speed curve when the train is running normally within the 101 s planned trip time.

It can be seen from Fig 11 that the first example of 15 s Later, the train will be informed to arrive to the next station 15 s later after running for 30-s. The results are shown in Fig 11 and Table 3. In Fig 11, as the current remaining trip time is extended from 71 s to 86 s, the train stops immediately accelerating and starts coasting. The train then continuously reduces its operating speed by braking. In addition, Table 3 summarizes the detailed performance of the $EITO_P$ after dynamically adjusting the trip time for the inter-station operation. The final running time of 15 s Later is 114 s, meeting the punctuality index (note that $T_p$ has been changed to 116 s). However, the sudden delay in the train's arrival time caused the train to decelerate in a larger $u$ at the beginning of the last speed-limited section of the interval. As a result, the passengers may feel discomfort for the increase in $I_c$ of $EITO_P$.

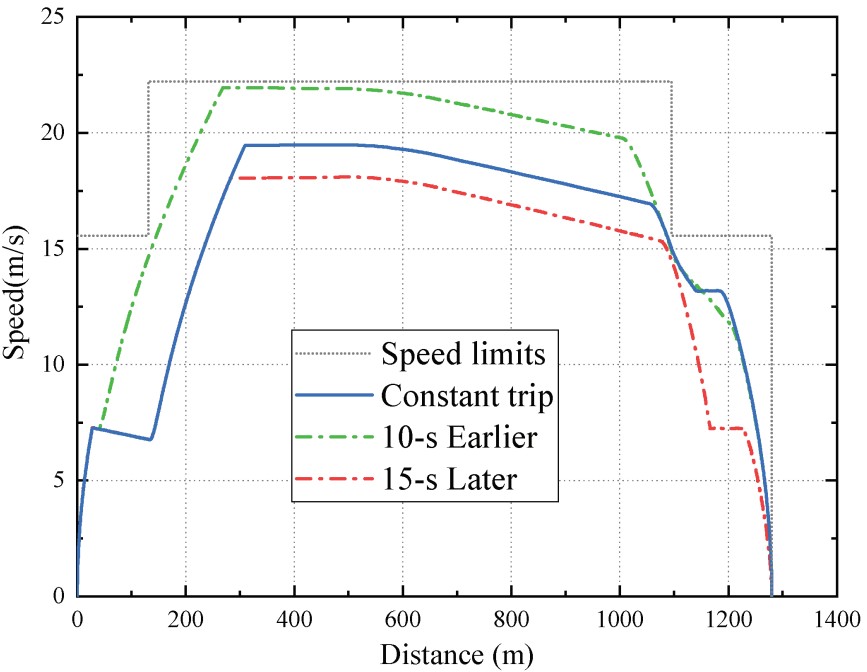

**Fig 11. EITO$_P$ with flexible trip time.**

**Table 3. EITO$_P$ with a variable trip time.**

| Performance | $t$ | $I_t$ | $I_s$ | $I_e$ | $I_c$ | $I_p$ |
|---|---|---|---|---|---|---|
| EITO$_p$(15 s Later) | 114 s | 1 | 1 | 315.9 | 2.97 | 1 |
| EITO$_p$(10 s Later) | 88 s | 1 | 1 | 459.6 | 4.10 | 1 |

The second example is the situation when the train is informed to arrive at the station 10 s earlier after running for 10 s, which is the converse of the first one. It means that the train's current remaining trip time suddenly reduces from 91 s to 81 s. By comparing the 10 s Earlier and Constant trip curves in Fig 11, we know that if the trip time decreases by 10 s due to an accident, EITO$_P$ will intelligently change its driving strategy and accelerate for the rest of the trip. Moreover, it can be known from Table 3 that the final running time is 88 s (note that $T_p$ has been changed to 91 s), which almost exceeds the requirement of the punctuality. It implies that despite the application of PPO and the improvement of the general performance of the metro operation, the punctuality of train operation is still affected by sudden changes in arrival times. Overall, EITO$_P$ can be flexible to cope with variable trip times.

Concerning unexpected speed limit changes: During the operation, we will simulate temporary speed restrictions (e.g., due to track maintenance). Preliminary results confirm that EITO$_P$ adjusts braking/coasting strategies in real time to comply with new limits while minimizing energy consumption. This new scenario will further validate the algorithm's effectiveness in handling unforeseen circumstances commonly encountered in real - world train operations, thus enhancing the reliability and practicality of our research findings.

In our study, we've already conducted tests on dynamic trip time adjustments, as presented in Case 2 of Sect 4.2. In these tests, EITO$_P$ has shown remarkable ability to adapt to sudden changes in the remaining journey time, as evidenced by Fig 11 and Table 3. Specifically, when

it comes to sudden time reduction, if the train is notified to arrive 10 seconds earlier midway, $EITO_P$ promptly responds by dynamically increasing the acceleration. As clearly shown in the "10 s Earlier" curve in Fig 11, this adjustment enables the train to meet the revised schedule. This vividly demonstrates $EITO_P$'s proficiency in quickly reacting to time - constrained scenarios. It can effectively optimize the train's operation, ensuring punctuality even under tight time pressures.

Regarding sudden time extension, when the train is informed that it can arrive 15 seconds later, $EITO_P$ takes appropriate action. As depicted in the "15 s Later" curve of Fig 11, the algorithm reduces the train's speed. By doing so, it manages to save energy while still maintaining punctuality. This not only showcases the adaptability of $EITO_P$ but also highlights its remarkable capacity to strike a balance between energy consumption and punctuality, two critical factors in train operation.

In conclusion, ITOR, STON, $EITO_E$, and $EITO_P$ can all generate reasonable control strategies and meet operating requirements for different planned trip times, thus demonstrating the flexibility of them. Besides, $EITO_P$ can dynamically adjust the train operation strategy in real-time by being informed of different arrival times (earlier or later arrival), indicating that $EITO_P$ also has a degree of intelligence.

## 4.3 Case 3

Considering the most current studies have tested models within a single interval, and few tested the robustness of algorithms in continuous line intervals with complex speed limits and gradients. Here, taking the continuous station interval from Songjiazhuang Station(SJZ) to Xiaocun Station(XC) and then from XC to Xiaohongmen Station(XHM) as an example to test the robustness of $EITO_E$ and $EITO_P$. As can be seen from Figs 4 and 12, the maximum gradient of Fig 12 (Case 3) is 500% of Fig 4 (Case 1), while the speed limits of the latter changes more dramatically.

Among them, the length from SJZ to XC is 2,631 m with a planned trip time of 190 s , and the length from XC to XHM is 1,274 m with a planned trip time of 108 s, i.e. the total trip time and total length are 290 s and 3,905 m. In this case, we ignore the stopping time, i.e., the train starts after arriving at the station and drives to the next station immediately.

Fig 13 shows the speed distance curves of EITO algorithms running from SJZ to XHM. We can see that the trajectories of the two curves are similar, and the maximum speed of $EITO_P$ is slightly lower than that of $EITO_E$, which indicates that $EITO_P$ may consume less energy than $EITO_E$. The average inference time for $EITO_P$ to generate a control action (acceleration/deceleration) at each time step (0.02 s) is 2.1 ms, which is 10× faster than the required control interval. This ensures real-time applicability even under strict operational deadlines.

In addition, when the train accelerates to the coasting point in the [1161 m, 2501 m] interval and starts coasting, the train gains positive acceleration. It means dangerous driving when accelerates to exceed the speed limit. The reason is that the interval has a downhill section with a gradients value of -0.008. However, due to the supervision of the Sect 3.1 safe speed $v_i^{\text{safe}}$, the train immediately adopts the maximum deceleration when the train is about to exceed the safe speed. This situation verifies that the proposed EITO algorithms can ensure the safety of train trips even with complicated speed limits and gradients.

The performance of EITO algorithms in complex continuous lines is shown in Table 4. The $I_c$ of both algorithms are larger due to the more lengthy and complex line conditions. And the train has to a stop and launch operation at the XC. However, both algorithms ensure that the train arrives at its destination safely and on time. The arrival times of $EITO_E$ to XC and XHM are 188 s and 112 s, respectively, while the arrival times of $EITO_P$ to XC and XHM

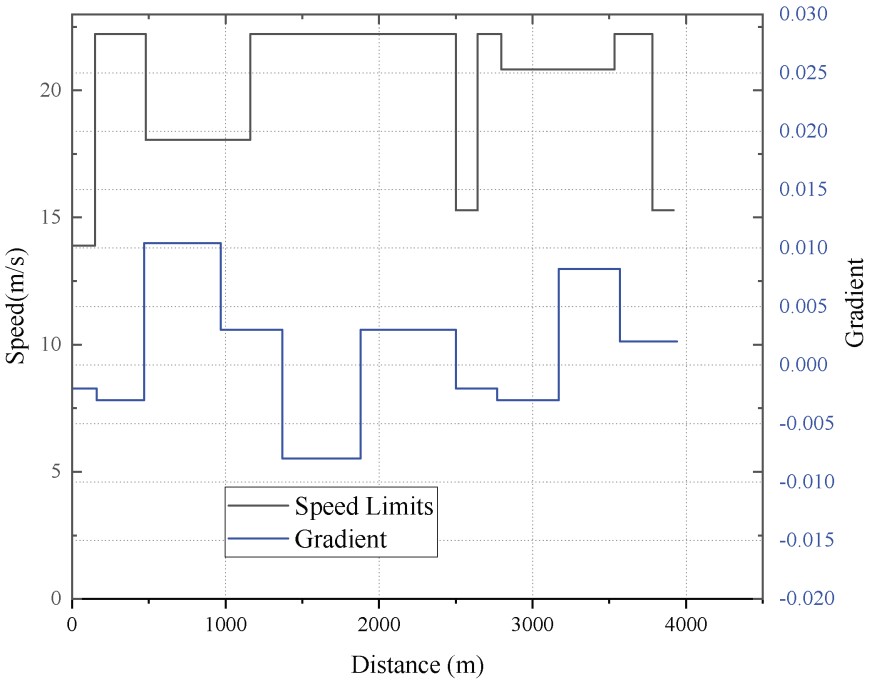

**Fig 12. Speed limits and gradient from SJZ to XHM.**

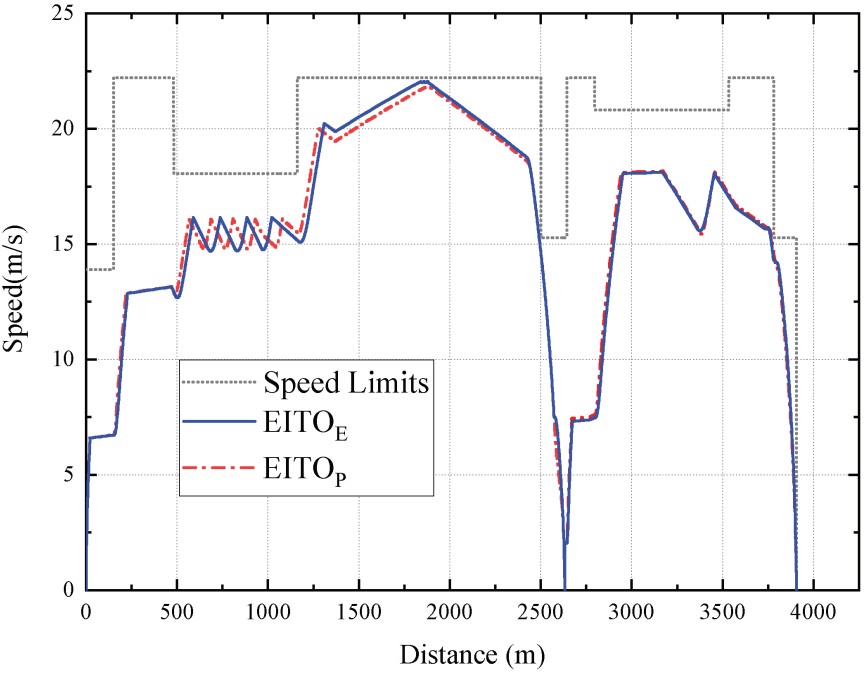

**Fig 13. Operation curve comparison among the EITO algorithms.**

**Table 4. Comparison of performance with complex continuous lines.**

| Performance | $t$ | $I_t$ | $I_s$ | $I_e$ | $I_c$ | $I_p$ |
|---|---|---|---|---|---|---|
| $\text{EITO}_E$(298 s) | 300 s | 1 | 1 | 1085.1 | 13.80 | 1 |
| $\text{EITO}_P$(298 s) | 298 s | 1 | 1 | 1048.9 | 12.17 | 1 |

are 189 s and 109 s, respectively, and the total travel times of the two algorithms are 300 s and 298 s, respectively. It can be seen that $\text{EITO}_P$ outperforms $\text{EITO}_E$ in terms of both inter-station and total travel on-time performance. Furthermore, $\text{EITO}_P$'s $I_c$ and $I_e$ are also lower than $\text{EITO}_E$. It also indicates that applying the PPO algorithm based on $\text{EITO}_E$ can effectively optimize the indicators of punctuality, energy-saving and comfort of $\text{EITO}_E$.

It can be seen from above analysis that energy-saving and comfort of both $\text{EITO}_E$ and $\text{EITO}_P$ have decreased moderately under the complex line conditions. However, both algorithms can ensure the train operate safely and punctually. This indicates that the proposed algorithms have good robustness. Dynamic Time Adjustment: Case 2 (Sect 4.2) demonstrates $\text{EITO}_p$'s ability to adapt to sudden trip time changes (e.g., $\pm 10$ – 15 s). The reward function (Eq 26) penalizes time deviations ($t_e \Delta t_e$), incentivizing the agent to adjust acceleration/coasting phases dynamically (Fig 11). For disturbance suppression (e.g., resistance uncertainty), $\text{EITO}_p$'s model-free PPO framework inherently adapts to unmodeled dynamics. We will include a dedicated robustness test (e.g., sudden resistance changes) in future work. The current experiments (Cases 1–3) validate $\text{EITO}_P$'s adaptability to: Variable trip times (95 s, 101 s, 115 s). Complex gradients and speed limits (Fig 12). Mid-journey schedule updates (Fig 11). These scenarios inherently cover "unpredictable conditions" by testing the algorithm's ability to replan trajectories in real time without prior offline profiles.

## 5 Conclusion

In this study, two EITO algorithms for intelligent train operation are proposed for addressing continuous metro operation control tasks, showcasing their ability to operate without the need for tracking offline speed profiles or relying on exact train model information. Our approach leverages an expert system to generate $\text{EITO}_E$ outputs based on driver experience and redefines key elements of the Proximal Policy Optimization (PPO) algorithm to develop $\text{EITO}_P$, which optimizes multiple operational objectives online. Through comparative analysis with existing intelligent driving algorithms and manual driving data, we demonstrated the superiority of our proposed algorithms in terms of safety, punctuality, energy efficiency, and passenger comfort. From the research and results of this work, several key conclusions can be drawn:

Flexible Intelligent Control: The EITO algorithms demonstrate the viability of intelligent train operations adaptable to real-time conditions, enhancing efficiency and adaptability beyond traditional, preset profiles.

Expert System Collaboration: Integrating expert system insights with $\text{EITO}_E$ significantly bolsters performance, underscoring the collaboration between human expertise and machine learning in intelligent train operations.

Multi-Objective Optimization: $\text{EITO}_P$ utilizes PPO and stands out as it comprehensively manages safety, punctuality, energy efficiency, and comfort, which are often conflicting objectives, through an overall control strategy.

Robustness in Complexity: Both $\text{EITO}_E$ and $\text{EITO}_P$ showcase robust performance in complex operational environments, with $\text{EITO}_P$ particularly adept at adjusting to varying trip times, crucial for real-world operational efficiency.

Energy and Comfort Superiority: Our algorithms surpass current methods in energy conservation and passenger comfort, tackling critical urban rail transit challenges and validating their practical application.

While our algorithms are promising, future work will focus on enhancing EITO$_P$'s dynamic adjustment capabilities and exploring cooperative control strategies for energy savings, including optimizing train schedules for regenerative energy utilization.

## Author contributions

**Conceptualization:** Yunhu Huang, Dewang Chen.

**Formal analysis:** Yunhu Huang.

**Investigation:** Yunhu Huang.

**Methodology:** Dewang Chen.

**Validation:** Jiateng Yin.

**Writing – original draft:** Yunhu Huang, Wenzhu Lai.

**Writing – review & editing:** Wenzhu Lai, Geng Lin, Jiateng Yin.

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
