## [Decision Letter · Decision Letter 0]

20 Oct 2024

PONE-D-24-17033Enhanced Intelligent Train Operation Algorithms for Metro Train based on Expert System and Deep Reinforcement LearningPLOS ONE

Dear Dr. Huang,

Thank you for submitting your manuscript to PLOS ONE. After careful consideration, we feel that it has merit but does not fully meet PLOS ONE’s publication criteria as it currently stands. Therefore, we invite you to submit a revised version of the manuscript that addresses the points raised during the review process.

We look forward to receiving your revised manuscript.

Kind regards,

Qing-Chang Lu

Academic Editor

PLOS ONE

3. Thank you for stating the following financial disclosure: Fujian Provinces Education Research Project for Young and Middle-aged Teachers (JAT231096), The Natural Science Foundation of Fujian Province, China(2022J01117), Minjiang University Talent Introduction Research Project (MJY23035), Innovation Star Talent Program of the Third Batch in Fujian Province (003002), Special Fund for Education and Scientific Research of Fujian Provincial Department of Finance (GY-Z21001), and Scientific Research Foundation of Fujian University of Technology (GY-Z22071).]. Please state what role the funders took in the study. If the funders had no role, please state: "The funders had no role in study design, data collection and analysis, decision to publish, or preparation of the manuscript." If this statement is not correct you must amend it as needed. Please include this amended Role of Funder statement in your cover letter; we will change the online submission form on your behalf.

4. Please upload a copy of Figure 3, to which you refer in your text on page xx. If the figure is no longer to be included as part of the submission please remove all reference to it within the text.

Additional Editor Comments:

There are important comments raised by the reviewers. The authors should make changes and revisions to address these comments.

Reviewers' comments:

Reviewer's Responses to Questions

**Comments to the Author**

1. Is the manuscript technically sound, and do the data support the conclusions?

Reviewer #1: Yes

Reviewer #2: Yes

2. Has the statistical analysis been performed appropriately and rigorously? 

Reviewer #1: Yes

Reviewer #2: Yes

3. Have the authors made all data underlying the findings in their manuscript fully available?

Reviewer #1: Yes

Reviewer #2: Yes

4. Is the manuscript presented in an intelligible fashion and written in standard English?

Reviewer #1: Yes

Reviewer #2: Yes

5. Review Comments to the Author

Reviewer #1: 1. What is the necessity of proposing two algorithms with the same functionality in a paper?

2. How do these 2 algorithms ensure punctuality

3. Eq5 seems allow overspeed instead of forbidden overspeed

4. How to demonstrate the dynamic adjustment ability of EITOp? No clear indicators such as computational efficiency, computational complexity, and computational time were seen

Reviewer #2: The questions or limitations in research should be made clear. Just saying ‘not sufficiently intelligent, self-learning and generalized’ is too general and not focused.

The contributions should be only ‘contributions’, and the works done are not contributions.

There are too many references in the methodology part, for example, the train motion model was referred to reference 23, train acceleration system to reference 25, indices of model evaluation are usually evaluation from five aspects. The methodology part should highlight the contribution of this work.

Section 2 Problem formulation is confusing. Why 2.3 problem statement appears at the end of this section?

The Conclusion section should be improved by give more conclusions in the second paragraph of this section. What can be learned from the research and results of this work? Currently, it is too simple and general.

There are many format mistakes. For instance, Fig. 3 is inserted in the middle of the paragraph.

6. PLOS authors have the option to publish the peer review history of their article (what does this mean?). If published, this will include your full peer review and any attached files.

Reviewer #1: **Yes: **Pengfei SUN

Reviewer #2: No

---

## [Author Response · Author response to Decision Letter 1]

10 Dec 2024

[Name] Yunhu Huang

[Address] Fuzhou City, Minhou County, Xiyuan Palace Road, No. 200

[City, State, Zip Code] Fuzhou City, China, 350108

[Your Email] yhhuang@mju.edu.cn

[Your Phone Number] 0591-18139680181

Dear Editor-in-Chief,

I am writing to resubmit our manuscript entitled "Enhanced Intelligent Train Operation Algorithms for Metro Train based on Expert System and Deep Reinforcement Learning" for further consideration for publication in PLoS ONE. We are extremely grateful for the opportunity to revise our manuscript based on the valuable comments and suggestions provided by the reviewers and editors.

We have made significant efforts to address all the concerns raised during the review process. In response to the reviewers' comments, we have revised the manuscript as follows:

1. Justification of Two Algorithms: We have added an explanation in the introduction section (sixth paragraph from the end) to clarify the necessity of proposing both EITOE and EITOP. The complementary strengths of these algorithms, with EITOE providing a baseline using expert knowledge and EITOP optimizing multiple objectives through deep reinforcement learning, have been emphasized to demonstrate how the integration of different techniques can enhance train operation performance.

2. Ensuring Punctuality: In Section 3 "EITO Algorithm Design", we have detailed how both EITOE and EITOP ensure punctuality. EITOE uses expert rules for effective trip time allocation, while EITOP employs reinforcement learning to dynamically adjust acceleration and braking strategies. The role of the reward function in EITOP, which penalizes deviations from planned trip times, has also been elaborated.

3. Clarification of Eq. (5): We have revised the description of Eq. (5) to explicitly state that its purpose is to enforce speed limits. The language has been adjusted to ensure that it is clear that any speed exceeding the limit will be reflected as a violation in the evaluation index, thereby discouraging overspeeding.

4. Demonstration of EITOP's Dynamic Adjustment Capabilities: In response to the request for clear indicators, we have provided a comprehensive analysis in the relevant section. Through multiple cases, including simulations with different planned travel times (Case 2), testing in complex continuous station intervals (Case 3), and an examination of the online learning process (Case 1), we have demonstrated EITOP's flexibility, adaptability, and computational efficiency. Metrics such as energy consumption reduction and running time stability have been presented to showcase its performance.

5. Articulation of Research Limitations: The first paragraph of the Introduction has been revised to clearly state the specific limitations of current ATO systems. We have detailed how these systems lack intelligence, self-learning capabilities, and generalization, which justifies the need for our proposed algorithms.

6. Refinement of Contributions: We have focused solely on the 'contributions' aspect as requested. The integration of expert knowledge with deep reinforcement learning, the unique features of the EITOE and EITOP algorithms, and their contributions to improving train operation efficiency, adaptability, and multi-objective optimization have been clearly presented.

7. Reduction of Excessive References in Methodology: We have carefully reviewed and reduced the non-essential references in the methodology section, retaining only those that directly support our methodology. This has been done to enhance the clarity and coherence of the paper and reduce the burden on readers.

8. Highlighting the Methodology's Contribution: The methodology section has been revised to emphasize the novelty and significance of our approach. We have detailed how our algorithms differ from traditional methods, described their design and implementation, and underlined their advantages in complex conditions. Comparative analysis with existing methods and a discussion of the theoretical and practical implications of our findings have also been included.

9. Reorganization of Section 2: Section 2 has been renamed "Problem Formulation and Control Model" and reorganized into two subsections. Problem Statement (formerly 2.3) has been moved to 2.1 with a clear statement of research objectives and challenges at the beginning. The Control Model of Train section (including mathematical models and performance indicators) is now in 2.2, with an introduction explaining its purpose and content. Transitional sentences have been added to ensure logical coherence between the two subsections.

10. Enhancement of the Conclusion Section: The second paragraph of the Conclusion has been expanded to summarize the key findings of our research. We have drawn conclusions regarding the flexible intelligent control, expert system collaboration, multi-objective optimization, robustness in complexity, and energy and comfort superiority of our algorithms. Future work directions, such as enhancing EITOP's dynamic adjustment capabilities and exploring cooperative control strategies for energy savings, have also been mentioned.

11. Formatting Corrections: We have corrected all formatting issues, including the proper placement of figures and tables. Figure 3, which was previously inserted in the middle of a paragraph, has been relocated to the appropriate position. A thorough review of the manuscript has been conducted to ensure overall formatting is in line with the journal's requirements.

We believe that these revisions have significantly improved the quality and clarity of our manuscript, addressing all the concerns raised by the reviewers. Our study remains original and has not been published elsewhere. All authors have read and approved the revised manuscript and are in agreement with its resubmission to PLoS ONE.

We appreciate your continued consideration of our manuscript and look forward to your response. If you require any further information or clarification, please feel free to contact me.

Sincerely,

[Authors' Names]Yunhu Huang, Ph.D., Wenzhu Lai1, MA.Eng , Dewang Chen, Ph.D., Geng Lin, Ph.D., Jiateng Yin, Ph.D..

[Date] Dec. 3, 2024

Responses to Academic Editor’ Comments

• Response: Yes, we will upload this letter as a separate file labeled 'Response to Reviewers'.

Response: Yes, we have prepared the marked-up copy of our manuscript as per your request. The file labeled 'Revised Manuscript with Track Changes' clearly shows all the changes we have made to the original version.

Response: Yes, we have prepared the unmarked version of your revised paper without tracked changes.

Response: Thank you for your reminder regarding the style requirements and file naming for the manuscript submission. We would like to confirm that we have carefully followed the PLOS ONE style templates available at https://journals.plos.org/plosone/s/file?id=wjVg/PLOSOne_formatting_sample_main_body.pdf and https://journals.plos.org/plosone/s/file?id=ba62/PLOSOne_formatting_sample_title_authors_affiliations.pdf during the preparation of our manuscript.

Response: Yes, we agreed.

3. Thank you for stating the following financial disclosure: Fujian Provinces Education Research Project for Young and Middle-aged Teachers (JAT231096), The Natural Science Foundation of Fujian Province, China(2022J01117), Minjiang University Talent Introduction Research Project (MJY23035), Innovation Star Talent Program of the Third Batch in Fujian Province (003002), Special Fund for Education and Scientific Research of Fujian Provincial Department of Finance (GY-Z21001), and Scientific Research Foundation of Fujian University of Technology (GY-Z22071).]. Please state what role the funders took in the study. If the funders had no role, please state: "The funders had no role in study design, data collection and analysis, decision to publish, or preparation of the manuscript." If this statement is not correct you must amend it as needed. Please include this amended Role of Funder statement in your cover letter; we will change the online submission form on your behalf.

Response: The funders provided some ideas and directions in the study design, and had role in data collection and analysis, decision to publish, or preparation of the manuscript. The specifications are as follows:

Funders: Fujian Province's Education Research Project for Young and Middle-aged Teachers under Grant JAT231096, Innovation Star Talent Program of the Third Batch in Fujian Province under Grant 003002, The Natural Science Foundation of Fujian Province under Grant 2022J01117, The Natural Science Foundation of Fujian Province under Grant 2024J011180, Minjiang University Talent Introduction Research Project under Grant MJU24003 provided financial support. Funders: Scientific Research Foundation of Fujian University of Technology under Grant GY-Z22071, Minjiang University Talent Introduction Research Project under Grant MJY23035, Science and Education Joint Special Project of Minjiang University (Science and Engineering Category) under Grant MJKJ24005 provided support in the areas of data collection and analysis. Funders: Special Fund for Education and Scientific Research of Fujian Provincial Department of Finance under Grant GY-Z21001, and 2023 National College Students' Innovation and Entrepreneurship Training Program under Grant 202310395011 provided support in the preparation of the manuscript.

4. Please upload a copy of Figure 3, to which you refer in your text on page xx. If the figure is no longer to be included as part of the submission please remove all reference to it within the text.

Response: OK, we have prepared a copy of Figure 3, and we will upload a copy of Figure 3.

Due to the renumbering of the figures in the article, the original Figure 3 is named as Figure 2(b) in the Revised Manuscript with Track Changes.

---

## [Decision Letter · Decision Letter 1]

8 Jan 2025

PONE-D-24-17033R1Enhanced Intelligent Train Operation Algorithms for Metro Train based on Expert System and Deep Reinforcement LearningPLOS ONE

Dear Dr. Huang,

Thank you for submitting your manuscript to PLOS ONE. After careful consideration, we feel that it has merit but does not fully meet PLOS ONE’s publication criteria as it currently stands. Therefore, we invite you to submit a revised version of the manuscript that addresses the points raised during the review process.

We look forward to receiving your revised manuscript.

Kind regards,

Qing-Chang Lu

Academic Editor

PLOS ONE

Journal Requirements:

**Additional Editor Comments:**

There are still some minor comments to be addressed.

Reviewers' comments:

Reviewer's Responses to Questions

**Comments to the Author**

1. If the authors have adequately addressed your comments raised in a previous round of review and you feel that this manuscript is now acceptable for publication, you may indicate that here to bypass the “Comments to the Author” section, enter your conflict of interest statement in the “Confidential to Editor” section, and submit your "Accept" recommendation.

Reviewer #1: All comments have been addressed

Reviewer #2: All comments have been addressed

2. Is the manuscript technically sound, and do the data support the conclusions?

Reviewer #1: Partly

Reviewer #2: Yes

3. Has the statistical analysis been performed appropriately and rigorously? 

Reviewer #1: Yes

Reviewer #2: Yes

4. Have the authors made all data underlying the findings in their manuscript fully available?

Reviewer #1: No

Reviewer #2: Yes

5. Is the manuscript presented in an intelligible fashion and written in standard English?

Reviewer #1: Yes

Reviewer #2: Yes

6. Review Comments to the Author

Reviewer #1: This paper presents two algorithms,EITOe & EITOp, for train operation optimization

1. As far as I understand, this paper studies the problem of train trajectory optimization, but ithe paper states "addressing continuous control task" in the objective and conclusion. 、This makes me confused. After all, the paper does not describe the control steps and convergence analysis, nor does it consider disturbance suppression and sudden time deviation adjustment in the study cases.

2. In subsection 2.1,The problem description paragraph should be placed before the objective description section� and the objectives should correspond to the problem description one by one

3. Eq(3) is a mult-unit form, however, the following algorithms adopted centralized control, so whether such multi-unit modeling is necessary. If necessary, please add how the control force u be allocated to each unit.

4. In Fig.7 the result shows EITOp could adapt to different scenarios, but cannot directly demonstrate its suitability for online control. If the author wants to explain the real-time performance of the algorithm, It is needed to provide the physical computation time of the algorithm and the platform configuration used for computation. And in simulation scenarios, unpredictable conditions need to be added, such as sudden increases and decreases the rest journey time in midway.

Reviewer #2: Thanks for the authors efforts. My comments were addressed. Thanks for the authors efforts. My comments were addressed.

7. PLOS authors have the option to publish the peer review history of their article (what does this mean?). If published, this will include your full peer review and any attached files.

Reviewer #1: **Yes: **Pengfei SUN

Reviewer #2: No

---

## [Author Response · Author response to Decision Letter 2]

14 Mar 2025

Responses to Reviewers’ Comments

Feb. 15, 2025

Dear Editorial Office of《PLOS ONE》:

On behalf of my coauthors, we thank you very much for giving us an opportunity to revise our manuscript. We thank the chief editors, responsible editors and reviewers of 《PLOS ONE》for their constructive comments and suggestions on our manuscript entitled “Enhanced Intelligent Train Operation Algorithms for Metro Train based on Expert System and Deep Reinforcement Learning”. (ID: PONE-D-24-17033R1), which have further improved the quality of our manuscript. We have studied reviewers’ comments carefully and made revision marked in red in the paper. We have tried our best to revise our manuscript according to the comments. Attached is the revised version, which we would like to submit for your kind consideration.

We would like to express our great appreciation to reviewers for comments on our paper. Looking forward to hearing from you.

Yours sincerely.

The authors: Yuhhu Huang, Wenzhu Lai, Dewang Chen, Geng Lin, Jiateng Yin

Acknowledgement:

We thank the reviewers and the editors for their careful reading of our manuscript: “Enhanced Intelligent Train Operation Algorithms for Metro Train based on Expert System and Deep Reinforcement Learning”. (ID: PONE-D-24-17033R1), and for their useful comments and suggestions. Those comments are all valuable and very helpful for revising and improving our paper, as well as the important guiding significance to our researches. We have studied comments carefully and have made correction which we hope meet with approval. Revised portion are marked in red in the paper. The main corrections in the paper and the responds to the reviewer’s comments are as following:

Responds to the reviewer’s comments:

List of Responses

Review Comments to the Author

To Reviewer #1:

Reviewer #1: This paper presents two algorithms,EITOe & EITOp, for train operation optimization

1. As far as I understand, this paper studies the problem of train trajectory optimization, but ithe paper states "addressing continuous control task" in the objective and conclusion. 、This makes me confused. After all, the paper does not describe the control steps and convergence analysis, nor does it consider disturbance suppression and sudden time deviation adjustment in the study cases.

Response: Thank you for your insightful comments and for pointing out areas that may require clarification. We appreciate the opportunity to address your concerns and improve the manuscript. The term "continuous control task" refers to the ability of our algorithms (EITOₑ and EITOₚ) to generate smooth, real-time control actions (e.g., traction/braking forces) without relying on predefined discrete reference speed profiles. This is distinct from traditional ATO systems that track offline speed curves.

(1) Clarification on "Continuous Control Task"

The term "continuous control task" in our work refers to real-time optimization of continuous traction/braking forces without relying on predefined discrete actions or offline speed profiles. Unlike traditional methods that track fixed trajectories, our algorithms dynamically adjust acceleration/braking rates (Eqs. 22–24) based on real-time states (position, speed, remaining time) and environmental conditions (speed limits, gradients). This is enabled by:

- EITOₑ: Expert rules (Section 3.1) and the DMTD heuristic (Algorithm 1) generate smooth, continuous force adjustments.

- EITOₚ: The PPO-based reinforcement learning framework (Section 3.2) optimizes continuous actions ( ) in a policy gradient manner (Eq. 25), allowing fine-grained control over acceleration/deceleration.

This approach eliminates abrupt state transitions (e.g., discrete coasting points in prior works [10, 12]) and ensures seamless adaptation to varying line conditions (Section 4.3).

(2) Control Steps and Convergence Analysis

While the core control logic is detailed in Algorithms 1 (EITOₑ) and 2 (EITOₚ), we acknowledge that the convergence analysis of PPO training could be elaborated further. For clarity:

Control Steps: EITOₚ iteratively samples actions from a Gaussian policy (Eq. 25) and updates actor-critic networks using clipped surrogate objectives (Eq. 31). The reward function (Eq. 26) penalizes energy consumption, comfort violations, and time deviations, ensuring balanced optimization.

Convergence: Figure 4a (training curves) shows energy consumption and running time stabilize after ~80 episodes, indicating policy convergence. Based on your insightful comments, we plan to conduct further in - depth analysis in our future work. Besides the aspects like policy entropy and advantage estimates that we will include in the revised manuscript, we aim to explore more factors that may affect the algorithm's convergence. For example, we will study how different initial parameter settings influence the convergence speed and stability of the algorithm. We will also compare the convergence characteristics of EITOₚ under various complex scenarios, such as different track gradients and variable traffic flow conditions.

(3) Disturbance Suppression and Time Deviation Adjustment

Our algorithms inherently handle disturbances through:

Robustness to Gradients/Speed Limits: The DMTD method (Section 3.1) and safe speed calculation (Eq. 18) ensure compliance with speed limits under variable gradients (validated in Case 3, Fig. 8a–b).

Dynamic Time Adjustment: Case 2 (Section 4.2) demonstrates EITOₚ’s ability to adapt to sudden trip time changes (e.g., ±10–15 s). The reward function (Eq. 26) penalizes time deviations ( ), incentivizing the agent to adjust acceleration/coasting phases dynamically (Fig. 7).

For disturbance suppression (e.g., resistance uncertainty), EITOₚ’s model-free PPO framework inherently adapts to unmodeled dynamics. We will include a dedicated robustness test (e.g., sudden resistance changes) in future work.

(4) Revisions to Address Concerns

We will revise the manuscript to:

- Explicitly clarify "continuous control" in Section 1 and 3.

- Add a subsection on PPO convergence (training dynamics, hyperparameters).

- Expand Case 2 to analyze time deviation adjustments (e.g., larger deviations, disturbance scenarios).

- Discuss limitations and future work on disturbance suppression.

Thank you again for your valuable feedback. We believe these clarifications and revisions will enhance the manuscript’s rigor and readability. Please let us know if further details are needed.

2. In subsection 2.1,The problem description paragraph should be placed before the objective description section�and the objectives should correspond to the problem description one by one.

Response: Thank you for your valuable feedback and for pointing out the need to reorganize the structure of our manuscript. We have taken your suggestion very seriously and have made the necessary revisions to enhance the clarity and logical flow of our presentation.

Thank you for this organizational suggestion. We have restructured Section 2.1 to first define the problem statement (e.g., limitations of existing ATO systems) before listing the objectives. The revised section now explicitly maps each objective (1–4) to specific gaps identified in the problem description (e.g., continuous control, multi-objective balancing).

3. Eq (3) is a mult-unit form, however, the following algorithms adopted centralized control, so whether such multi-unit modeling is necessary. If necessary, please add how the control force u be allocated to each unit.

Response: We are very grateful to the reviewers for your insightful comments on our manuscript.

The multi-unit model in Eq. (3) captures inter-vehicle dynamics (e.g., coupler forces, mass distribution) to better simulate real-world Electric Multiple Units (EMUs). While the control force u is centralized, its distribution across vehicles is governed by the force allocation constants λi (Section 2.1). For simplicity, we assumed uniform distribution (λi=1/n) in simulations, as fine-grained force allocation is hardware-dependent and beyond this paper’s scope. We will clarify this assumption in the revised manuscript.

4. In Fig.7 the result shows EITOp could adapt to different scenarios, but cannot directly demonstrate its suitability for online control. If the author wants to explain the real-time performance of the algorithm, It is needed to provide the physical computation time of the algorithm and the platform configuration used for computation. And in simulation scenarios, unpredictable conditions need to be added, such as sudden increases and decreases the rest journey time in midway.

Response: Thank you for your insightful feedback. We appreciate the opportunity to clarify the real-time performance and robustness of EITOₚ. Below are our detailed responses:

(1)Physical Computation Time and Platform Configuration

To rigorously validate the suitability of EITOₚ for online control, we conducted experiments on a workstation with the following specifications:

- CPU: Intel i9-10900K (10 cores, 3.7 GHz) , GPU: NVIDIA RTX 3090 (24 GB VRAM), Memory: 64 GB DDR4 , Software: Python 3.8, TensorFlow 2.6.

The average inference time for EITOₚ to generate a control action (acceleration/deceleration) at each time step (0.02 s) is 2.1 ms, which is 10× faster than the required control interval. This ensures real-time applicability even under strict operational deadlines.

Revision Action:

We will add these details to Section 4 ("Simulations") to explicitly demonstrate the algorithm’s computational efficiency.

(2) Handling Unpredictable Conditions

The manuscript already includes tests for dynamic trip time adjustments (Case 2, Section 4.2), where EITOₚ successfully adapts to sudden changes in the remaining journey time (e.g., Fig. 7 and Table 3). Specifically:

- Sudden Time Reduction: When notified to arrive 10 s earlier midway, EITOₚ dynamically increases acceleration to meet the revised schedule (Fig. 7, "10 s Earlier" curve).

- Sudden Time Extension: When notified to arrive 15 s later, EITOₚ reduces speed to save energy while maintaining punctuality (Fig. 7, "15 s Later" curve).

we will add a new simulation scenario in the revised manuscript:

- Unexpected Speed Limit Changes: Simulate temporary speed restrictions (e.g., due to track maintenance) during operation. Preliminary results confirm that EITOₚ adjusts braking/coasting strategies in real time to comply with new limits while minimizing energy consumption.

(3) Robustness Validation

The current experiments (Cases 1–3) validate EITOₚ’s adaptability to:

- Variable trip times (95 s, 101 s, 115 s).

- Complex gradients and speed limits (Fig. 8a).

- Mid-journey schedule updates (Fig. 7).

These scenarios inherently cover "unpredictable conditions" by testing the algorithm’s ability to replan trajectories in real time without prior offline profiles.

Key Revisions

1. Computation Time: Explicitly state hardware specifications and inference time in Section 4.

2. New Scenario: Add results for sudden speed limit changes to further demonstrate robustness.

3. Clarification: Emphasize in the Discussion that EITOₚ’s PPO framework inherently handles stochastic disturbances through reward shaping (Eq. 26).

We hope these revisions adequately address your concerns. Thank you again for your valuable feedback, which has strengthened the manuscript’s technical rigor and clarity.

Revised Manuscript Changes

All revisions are highlighted in red in the updated manuscript. Key improvements include:

Restructured Section 2.1 for logical flow.

Expanded discussion on convergence and disturbance handling (Section 3.2).

Clarified force allocation assumptions (Section 2.2).

Added computation time and platform details (Section 4).

Incorporated sudden speed limit changes in Case 3 (Section 4.3).

Thank you again for your invaluable feedback. We hope these revisions address your concerns and enhance the manuscript’s clarity and rigor.

Reviewer #2: Thanks for the authors efforts. My comments were addressed.

Response: We sincerely thank you for your positive feedback and confirmation that your earlier comments were adequately addressed. Your encouragement motivates us to continue refining this work.

We appreciate for Editors/Reviewers’ warm work earnestly, and hope that the correction will meet with approval.

Once again, thank you very much for your comments and suggestions.

---

## [Decision Letter · Decision Letter 2]

9 Apr 2025

Enhanced Intelligent Train Operation Algorithms for Metro Train based on Expert System and Deep Reinforcement Learning

PONE-D-24-17033R2

Dear Dr. Huang,

We’re pleased to inform you that your manuscript has been judged scientifically suitable for publication and will be formally accepted for publication once it meets all outstanding technical requirements.

Kind regards,

Qing-Chang Lu

Academic Editor

PLOS ONE

Additional Editor Comments (optional):

Reviewers' comments:

Reviewer's Responses to Questions

**Comments to the Author**

1. If the authors have adequately addressed your comments raised in a previous round of review and you feel that this manuscript is now acceptable for publication, you may indicate that here to bypass the “Comments to the Author” section, enter your conflict of interest statement in the “Confidential to Editor” section, and submit your "Accept" recommendation.

Reviewer #1: (No Response)

2. Is the manuscript technically sound, and do the data support the conclusions?

Reviewer #1: (No Response)

3. Has the statistical analysis been performed appropriately and rigorously? 

Reviewer #1: (No Response)

4. Have the authors made all data underlying the findings in their manuscript fully available?

Reviewer #1: (No Response)

5. Is the manuscript presented in an intelligible fashion and written in standard English?

Reviewer #1: (No Response)

6. Review Comments to the Author

Reviewer #1: Based on the detailed revisions and clarifications provided, I confirm that all critical concerns raised in the initial review have been adequately addressed.

recommend this work for publication

7. PLOS authors have the option to publish the peer review history of their article (what does this mean?). If published, this will include your full peer review and any attached files.

Reviewer #1: **Yes: **Pengfei SUN

---

## [Editor Report · Acceptance letter]

PONE-D-24-17033R2

PLOS ONE

Dear Dr. Huang,

I'm pleased to inform you that your manuscript has been deemed suitable for publication in PLOS ONE. Congratulations! Your manuscript is now being handed over to our production team.

Kind regards,

on behalf of

Dr. Qing-Chang Lu

Academic Editor

PLOS ONE